# Bayesian atmospheric tomography for detection and quantification of methane emissions: Application to data from the 2015 Ginninderra release experiment

Laura Cartwright[1], Andrew Zammit-Mangion[1], Sangeeta Bhatia[2, 3], Ivan Schroder[4], Frances Phillips[5], Trevor Coates[6], Karita Negandhi[7, 8], Travis Naylor[5], Martin Kennedy[7], Steve Zegelin[9], Nick Wokker[10], Nicholas M. Deutscher[5], and Andrew Feitz[4]

[1]School of Mathematics and Applied Statistics, University of Wollongong, Wollongong, Australia
[2]Centre for Research in Mathematics, Western Sydney University, Parramatta, Australia
[3]School of Public Health, Imperial College London, London, UK (current address)
[4]Geoscience Australia, Canberra, Australia
[5]Centre for Atmospheric Chemistry, School of Earth, Atmosphere and Life Sciences, University of Wollongong, Wollongong, Australia
[6]School of Agriculture and Food, University of Melbourne, Melbourne, Australia
[7]Department of Earth and Planetary Sciences, Macquarie University, Sydney, Australia
[8]Office of Environment and Heritage, Parramatta, Australia (current address)
[9]CSIRO Oceans and Atmosphere, Canberra, Australia
[10]Department of Industry, Innovation and Science, Canberra, Australia

**Correspondence:** Laura Cartwright (lcartwri@uow.edu.au)

**Abstract.** Detection and quantification of greenhouse-gas emissions is important for both compliance and environment conservation. However, despite several decades of active research, it remains predominantly an open problem, largely due to model errors and assumptions that appear at each stage of the inversion processing chain. In 2015, a controlled-release experiment headed by Geoscience Australia was carried out at the Ginninderra Controlled Release Facility, and a variety of instruments and methods were employed for quantifying the release rates of methane and carbon dioxide from a point source. This paper proposes a fully Bayesian approach to atmospheric tomography for inferring the methane emission rate of this point source using data collected during the experiment from both point- and path-sampling instruments. The Bayesian framework is designed to account for uncertainty in the parametrisations of measurements, the meteorological data, and the atmospheric model itself when doing inversion using Markov chain Monte Carlo (MCMC). We apply our framework to all instrument groups using measurements from two release-rate periods. We show that the inversion framework is robust to instrument type and meteorological conditions. From all the inversions we conducted across the different instrument groups and release-rate periods, our worst-case median emission rate estimate was within 36% of the true emission rate. Further, in the worst case, the closest limit of the 95% credible interval to the true emission rate was within 11% of this true value.

# 1 Introduction

Methane ($CH_4$) is an important transition fuel for decarbonisation of the global energy system (International Energy Agency, 2017). As countries increase the renewable energy mix into their existing electricity networks, $CH_4$ can firm up network stability and supply (International Energy Agency, 2017; Jenkins et al., 2018). Utilisation of biogas or natural gas with carbon

capture and storage offers a lower cost pathway to achieve deep decarbonisation targets (Sepulveda et al., 2018). One of the disadvantages of $CH_4$, however, is that its global warming potential is much greater than that of carbon dioxide ($CO_2$), so that only a few percent of losses of $CH_4$ into the atmosphere can negate any climate-change mitigation advantages from reducing conventional coal-fired power production (Kinnon et al., 2018). For this reason, it is critical that losses of $CH_4$ along the supply chain are accurately accounted for to ensure public confidence in climate-change mitigation benefits of switching to natural gas.

Unfortunately, while several types of instrumentation are available to aid the detection and estimation of fugitive emissions, harnessing acquired data for reliable emission detection and quantification remains a notoriously difficult problem.

Several controlled release experiments of $CH_4$ and $CO_2$ have been conducted in order to improve techniques for estimating greenhouse gas emissions (Flesch et al., 2004; Lewicki and Hilley, 2009; Loh et al., 2009; Etheridge et al., 2011; Humphries et al., 2012; van Leeuwen et al., 2013; Luhar et al., 2014; Jenkins et al., 2016; Ars et al., 2017). Building on this body of work,

in 2015 a $CH_4$ and $CO_2$ controlled-release experiment was held at the Ginninderra Controlled Release Facility in Canberra, Australia (Feitz et al., 2018). This large multidisciplinary, multi-institutional blind-release trial (i.e., the participants did not know the true release rate) simultaneously assessed eight different $CH_4$ emission-rate estimation techniques, using data from both mobile and stationary instrumentation. These eight techniques included tracer ratio techniques; backwards Lagrangian stochastic modelling; forward Lagrangian stochastic modelling; Lagrangian stochastic footprint modelling; and atmospheric

tomography techniques. A full description of the methods and results is given in Feitz et al. (2018).

Every group involved in the analysis presented in Feitz et al. (2018) used a unique combination of instrumentation and estimation technique when carrying out the analysis, making it hard to establish the respective merits (or otherwise) of the employed techniques from the inversion results. Nonetheless, an interesting observation from the study is that none of the eight techniques deployed during the blind release trial had a leakage uncertainty range (95% interval) that included the true

emission rate, while some estimates (including one obtained using atmospheric tomography) were factors of 2 or more off from the true value. Given that atmospheric methane concentration and meteorological instrument measurement uncertainty is generally low for each of the different approaches, it suggests that the techniques that were used did not adequately account for the variability of atmospheric measurements or the uncertainty introduced through parametrisation of atmospheric mixing conditions (e.g., Monin-Obukhov lengths and/or Pasquill stability classes; see Sect. 3.1) and atmospheric dispersion/transport

model uncertainty.

A number of studies have highlighted the importance of atmospheric-model error in estimating emission rates or fluxes (e.g., Chevallier et al., 2010; Basu et al., 2018). For example, Peylin et al. (2002) showed that flux estimates are sensitive to the chosen spatio-temporal resolution of the fluxes and the chosen transport model. Uncertainty in the meteorological fields driving the transport model is also known to play a big role (e.g., Miller et al., 2015). While ensemble inversions are frequently used to

highlight the sensitivity of the results to atmospheric models and meteorological fields, learning unknown parameters associated with transport *concurrently* with the emission rate is not often done. This is largely due to the computational implications of such an approach. Key here are the use of surrogate models (or emulators) to obtain simplified transport representations. For example, Lucas et al. (2017) use decision/regression trees as a surrogate for FLEXPART-WRF, which allows for quick simulation at various parameter settings that can in turn be used to make inference. For the Ginninderra data we employ the more traditional Gaussian plume model, which can be seen as a surrogate for a full-blown transport model. While known to work well in the small domain (an area of approximately $100 \times 100$ m) setting we consider (e.g., Riddick et al., 2017), importantly this plume model is quick to simulate from, giving us the opportunity to calibrate it while estimating the emission release rate (e.g., Borysiewicz et al., 2012). As we see in our sensitivity analysis of our results in Sect. 6, "online plume-model calibration" is crucial for obtaining accurate emission-rate estimates with our data.

The transport model plays an important role in inverse modelling. Calibration of the transport model from observations can be done within the classic inverse theory framework of Tarantola (2005). This framework is in turn seated within a Bayesian paradigm, which underpins several of the inversion systems in place today; see, for example, Flesch et al. (2004); Humphries et al. (2012); Hirst et al. (2013); Ganesan et al. (2014); Luhar et al. (2014); Houweling et al. (2017); White et al. (2018). Inference in such cases is often done using sampling techniques such as Markov chain Monte Carlo (MCMC) or importance sampling (Rajaona et al., 2015). Quick evaluation of the transport/dispersion model (or surrogate) is crucial when repeatedly evaluating it within an MCMC framework; the Gaussian plume model is hence a popular choice in these frameworks (e.g., Jones et al., 2016; Wang et al., 2017). MCMC is also our method of choice for Bayesian atmospheric tomography, because it allows relatively easy computation of posterior distributions of parameters that are deeply nested within a hierarchical model. It is also ideally suited for the case of point-source emissions, where the dimensionality of the latent space is low (unlike, for example, when doing regional emission quantification).

Atmospheric tomography, a term inspired from medical imaging, combines data from a collection of measurement sites with Bayesian inversion to detect and quantify emissions. The primary contribution of this article is an extension of the atmospheric tomography technique described in Sect. 2.4.2 of Feitz et al. (2018). In Feitz et al. (2018), atmospheric tomography was only used on one type of instrument and did not account for uncertainty in the transport model. The technique we propose accounts for uncertainty in our data, in our process models, and in our parameters; is applicable to both point and path-sampling instruments; and takes into account instrument-specific bias. Inference is made on all unknown parameters using MCMC and uncertainty in the transport-model parameters are propagated to our posterior inferences on the release rate. We demonstrate the efficacy and utility of the unifying Bayesian framework on data from point- and path-referenced instruments used in the Ginninderra experiment. A secondary contribution is the curated provision of a data set containing a large portion of the Ginninderra data at a five-minute resolution, which we hope will serve as a resource for other researchers to validate their own emission-rate estimation techniques on. The data and scripts required to reproduce the results in this article are available from https://github.com/Lcartwright94/BayesianAT.

The remainder of the article is organised as follows. Section 2 gives an overview of the experimental setup and the data collected during the 2015 Ginninderra experiment. Section 3 describes the atmospheric transport model used, while Sect. 4

details the hierarchical model we employ and the Bayesian methodology we develop for emission-rate estimation. Section 5 gives the results from application of our Bayesian atmospheric tomography technique on the Ginninderra data. Section 6 examines how our results would change if certain components in our model (e.g., relating to the plume model) are (erroneously) assumed fixed and known. Section 7 concludes.

## 2   The 2015 Ginninderra release experiment

A full description of the experimental setup, measurement techniques and quantification methods used in the 2015 Ginninderra release experiment are given in Feitz et al. (2018). Briefly, $CH_4$ (together with $CO_2$ and nitrous oxide), was released from a small chamber located in a fallow agricultural field from 23 April to 12 June 2015, and 23 to 24 June 2015. A variety of $CH_4$ sensors were placed around the release chamber. The measurement data considered in this study were obtained from two Picarro G2201-i analysers (positioned in the predominant upwind (NW) and downwind (SE) location of the release chamber, labelled Picarro.West and Picarro.East, respectively), four eddy covariance (EC) towers equipped with Li-COR 7700 open path $CH_4$ sensors (labelled EC.A, EC.C, EC.D, and EC.E, respectively), two scanning FTIR analysers with four retro-reflectors terminating six measurement paths (labelled P1 to P6, respectively), and a scanning GasFinder 2 Boreal laser with seven reflectors forming seven measurement paths (labelled R1 to R7, respectively); see the left panel of Fig. 1. Meteorological data was collected from EC.A equipped with a Vaisala HMP50 relative humidity and temperature sensor, a CSI EC150 $CO_2$-$H_2O$ sensor, a Li-COR 7700 $CH_4$ sensor, a Kipp and Zonen CNR4 radiometer, a CSI CSAT3 sonic anemometer, and a Gill Wind-Sonic anemometer. Wind speed and wind direction were measured by the CSAT3 sonic anemometer and the Gill WindSonic anemometer. As part of data quality control, horizontal wind speed and wind direction data from the two instruments were compared, with no arising issues. Both sonic anemometers were using factory calibration. Wind directions were determined by manually aligning the sonic anemometers so that the reference direction was true north. Data from CSAT3 sonic anemometer was logged at 10 Hz, and data from the Gill WindSonic anemometer at 1 Hz.

The gases were released at a height of 0.3 m, and the standard $CH_4$ release rate was 5.8 g min$^{-1}$, limited mostly to daylight hours. On brief occasions, the $CH_4$ release rate varied between 2.9 and 20 g min$^{-1}$ to enable testing of mobile $CH_4$ sensor platforms. Towards the end of the experiment (8 to 12 June), the $CH_4$ release rate was decreased from 5.8 to 5.0 g min$^{-1}$ and the setup for the Boreal laser measurements was modified with the number of retro-reflectors and paths reduced to six (labelled R8 to R13, respectively; see the right panel of Fig. 1). The location of all other $CH_4$ sensors did not change over the duration of the experiment. The Picarro analysers were not deployed until 21 May, and the $CH_4$ release rate on 23 and 24 June was constantly varied. Hence, in this article we only consider data between 21 May and 12 June, excluding 26 and 27 May where the release rate was also constantly varied.

The data set used to obtain the results presented in Sect. 5 was compiled by pooling together the separate meteorological and concentration data sets used in the Ginninderra experiment. A common resolution of five minutes was chosen, that is, all measurements of concentration and meteorological variables were averaged over a regular set of five-minute intervals. Measured $CH_4$ concentrations were then matched with corresponding meteorological measurements by time and placed into

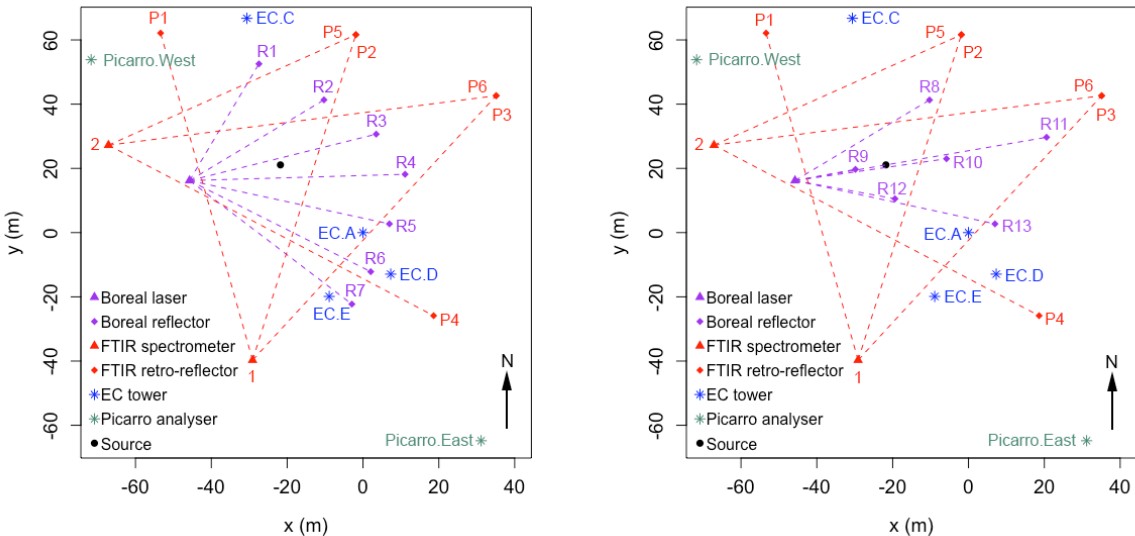

**Figure 1.** Left: Layout of instruments in the 2015 Ginninderra release experiment between 21 May and 7 June 2015. Right: Layout of instruments between 8 and 12 June 2015. R1 to R13 are the paths formed between the Boreal laser and reflectors; P1 to P6 are the paths formed between the FTIR spectrometers and retro-reflectors; EC.A to EC.E are the EC towers; and Picarro.East and Picarro.West are the Picarro analysers. All coordinates are relative to EC.A, which is situated at the origin.

long-table format, with each row corresponding to a unique data point. For path measurements, two extra columns were used to denote the end-point coordinates of the paths.

Initial pre-processing was carried out to provide a complete data set without outliers. First, data containing missing values considered critical for emission-rate estimation (in particular, air temperature, air pressure, wind speed, and wind direction) were removed from the data set. Second, data points corresponding to upwind measurements that were more than three median absolute deviations away from the instrument's median upwind measured concentration, were determined to be outliers, and hence removed. A point measurement was classified as upwind if the angle subtended from the source by a line joining the instrument location to the plume centreline was more than $45°$. A path measurement was classified as upwind if the angles subtended at every point along the path were more than $45°$.

## 3   Transport modelling

In this section we detail the plume model employed, and how it is used to supply model-predicted concentrations for the path measurements.

**Table 1.** Stability classes to which observations within the Ginninderra experiment are allocated, and the corresponding values of $a_{k_i}, b_{k_i}, c_{k_i}$, and $d_{k_i}$ used to construct the horizontal ($\sigma_{y_i,k_i}$), and vertical ($\sigma_{z_i,k_i}$) standard deviations of the plume when $x_i$ is in m.

| Stability Class ($k_i$) | Stability Condition | $a_{k_i}$ | $b_{k_i}$ | $c_{k_i}$ | $d_{k_i}$ |
|---|---|---|---|---|---|
| A | Extremely unstable | 0.17993 | 0.94470 | 24.167 | 2.5334 |
| B | Moderately unstable | 0.14506 | 0.93198 | 18.333 | 1.8096 |
| C | Slightly unstable | 0.11025 | 0.91465 | 12.500 | 1.0857 |
| D | Neutral | 0.084739 | 0.86974 | 8.3330 | 0.72382 |
| E | Slightly stable | 0.075005 | 0.83660 | 6.2500 | 0.54287 |
| F | Moderately stable | 0.054370 | 0.81558 | 4.1667 | 0.36191 |

## 3.1 Gaussian plume dispersion modelling

As outlined in Sect. 1, we use a transport model that is simply parameterised, and easy to evaluate, so that it can be calibrated online. One of the simplest models that works well on the short distances we consider is the Gaussian plume dispersion model (e.g., Wark et al., 1998, Chapter 4). Here the true emission rate is denoted by $Q$ in g s$^{-1}$, the height of the CH$_4$ point source 5 by $H$ in m, and the total number of observations by $N$. The classic Gaussian plume model is given by

$$C(x_i, y_i, z_i, Q, U_i, H, \boldsymbol{\theta}_{k_i}) = \frac{Q}{2\pi U_i \sigma_{y_i,k_i} \sigma_{z_i,k_i}} \exp\left(-\frac{y_i^2}{2\sigma_{y_i,k_i}^2}\right) \left[\exp\left(-\frac{(z_i - H)^2}{2\sigma_{z_i,k_i}^2}\right) + \exp\left(-\frac{(z_i + H)^2}{2\sigma_{z_i,k_i}^2}\right)\right], \tag{1}$$

where $C$ is the *model-predicted* concentration in g m$^{-3}$ of CH$_4$ at a single spatial point $(x_i, y_i, z_i)$ in m along the direction of the plume corresponding to the $i$th measurement; $U_i$ is the wind speed associated with the $i$th measurement in m s$^{-1}$; $k_i \in \{A, B, C, D, E, F\}$ represents the Pasquill stability class (a categorisation reflective of the expected level of horizontal 10 and/or vertical spread of the atmospheric particles after emission; see Pasquill, 1961) associated with the $i$th measurement; and $\boldsymbol{\theta}_{k_i}$ are plume-specific parameters used the construct the standard deviations $\sigma_{z_i,k_i}$ and $\sigma_{y_i,k_i}$. These standard deviations of the plume in the vertical and horizontal directions are given by

$$\sigma_{z_i,k_i} = a_{k_i} x_i^{b_{k_i}},$$

$$\sigma_{y_i,k_i} = 0.4651 x_i \tan(\nu_i),$$

15 respectively, where $\nu_i = 0.01745 \left(c_{k_i} - d_{k_i} \ln(x_i/1000)\right)$. Note that the coefficients $a_{k_i}, b_{k_i}, c_{k_i}, d_{k_i}$ correspond to the $i$th measurement and depend on the stability class associated with that measurement, $k_i \in \{A, B, C, D, E, F\}$. Values for these coefficients by stability class are given in Wark et al. (1998, Chapter 4) and shown here in Table 1 for completeness. We collect the plume-specific parameters in $\boldsymbol{\theta}_{k_i} \equiv (a_{k_i}, b_{k_i}, c_{k_i}, d_{k_i})'$ where $'$ denotes the transpose operator.

The stability class to which an observation is allocated is classically based on (i) the Monin-Obukhov length (the theoretical 20 height at which turbulence is produced by buoyancy and mechanical forces in equal amounts; see Sienfeld and Pandis (2006),

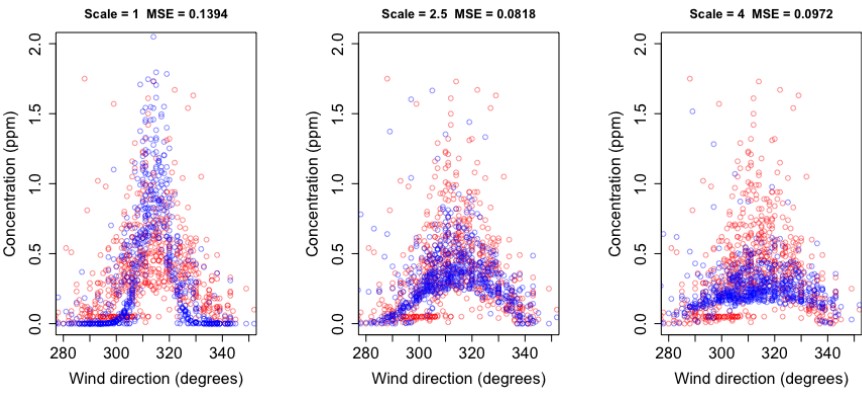

**Figure 2.** Predicted (blue) and observed (red) enhancements in ppm at EC.A between May 21 and June 7 2015 when scaling $\sigma_{y_i, k_i}$ by 1, 2.5, and 4, respectively. The mean-squared errors (MSE) between the observed and the predicted enhancements are also shown. Of the three, the best agreement between predicted and observed values occurs when $\sigma_{y_i, k_i}$ is scaled by 2.5.

Chapter 16), and (ii) an effective roughness length. The Monin-Obukhov length ($L$-value) is given by $L = -u_*^3 \bar{\xi}_v / (qg(\overline{w^* \xi_v^*})_s)$ (Jacobson, 2005, Chapter 8), where $u_*$ is the frictional velocity, $\bar{\xi}_v$ is the mean virtual potential temperature, $(\overline{w^* \xi_v^*})_s$ is the surface virtual potential temperature flux, $q$ is the von Kármán constant, and $g$ is the acceleration due to gravity. In our case we used WindTrax (http://www.thunderbeachscientific.com/windtrax.html) to determine the $L$-value for each observation; we provide the $L$-values with the compiled data. We set the effective roughness length $z_0 = 0.01$ m, corresponding to a relatively flat area, with short or no grass, and minimal buildings/trees/other obstacles; see Sienfeld and Pandis (2006, Chapter 16) and World Meteorological Organisation (2008, Chapter 5). This is a suitable choice for the Ginninderra site. We used the results of Golder (1972) to allocate a stability class to each observation based on the $L$-values provided by WindTrax and $z_0 = 0.01$ m.

The coefficients typically used for each stability class could be off by a factor of 2 or more (Wark et al., 1998, Chapter 4). To show that this is also the case with our categorisation scheme, in Fig. 2 we show the Gaussian-plume-model predicted outputs together with the observed data enhancements (see Sect. 4.1) at one of our measurement locations (namely, EC.A) between 21 May and 7 June, when scaling $\sigma_{y_i, k_i}$ by 1, 2.5, and 4, respectively. Clearly, with no scaling the predicted plume is too narrow, while with a scaling of 4 is too broad. A scaling of 2.5 gives good agreement. Importantly, since in Eq. (1) $Q$ only serves to scale the predicted concentrations (i.e., make them larger or smaller by a constant factor), it is apparent that this plume scaling factor is identifiable, in the sense that we can learn it from the data *while* estimating the emission rate (provided the source is active). Online plume-model calibration fits naturally within the MCMC framework discussed in Sect. 4.

### 3.2 Low wind speeds

It is well known that the Gaussian plume model is less accurate for low wind speeds (e.g., Turner, 1994, Chapter 2). One reason for this is that the wind-speed $U_i$ is in the denominator of the scaling coefficient of Eq. (1); hence, the plume model prediction becomes very sensitive to $U_i$ as it tends towards zero. This is problematic as $U_i$, although often assumed known,

is an average calculated from noisy measurements taken over some time span (in our case five minutes), and is thus itself noisy. From Eq. (1) we see that, when conditioned on all other parameters, the variance of $C_i$ is proportional to the variance of the inverse of $U_i$, which can be very large for small $U_i$. Instead of removing data at low wind speeds as is often done (e.g., Feitz et al., 2018), we analyse the theoretical relationship between the variance of the inverse wind speed and $C_i$. We then use

this relationship to discount low wind-speed model-predictions in the Bayesian framework in a principled manner. While the analyst still needs to choose a cutoff below which to model this relationship, in separate studies we found that our inferences are not particularly sensitive to the chosen cutoff. Moreover, we found that downweighting instead of excluding was necessary for making inference when not many observations associated with high wind speeds were available.

Each wind speed $U_i$ is an average of a number of wind speeds (say $n_{U_i}$) recorded over five minutes. Therefore $U_i$ is a

sample mean, and thus an unbiased estimator of the true (population) mean wind speed, say $\mu_{U_i}$, over this time interval. By the Central Limit Theorem, $\sqrt{n_{U_i}}(U_i - \mu_{U_i}) \xrightarrow{\mathcal{D}} \text{Gau}(0, \sigma_{U_i}^2)$, where $\mathcal{D}$ implies convergence in distribution, $\text{Gau}(\mu, \sigma^2)$ denotes the Gaussian distribution with mean $\mu$ and variance $\sigma^2$, and $\sigma_{U_i}^2$ is the variance of the wind speeds over the $i$th time interval, which was derived from the raw (disaggregated) data. We can then use the delta method (e.g., Casella and Berger, 2002, Chapter 5) to deduce that

$$\sqrt{n_{U_i}}\left(\frac{1}{U_i} - \frac{1}{\mu_{U_i}}\right) \xrightarrow{\mathcal{D}} \text{Gau}\left(0, \left(\frac{\mathrm{d}}{\mathrm{d}\mu_{U_i}}\left(\frac{1}{\mu_{U_i}}\right)\right)^2 \sigma_{U_i}^2\right).$$

Hence, the variance of $1/U_i$ is approximately

$$\left(\frac{\mathrm{d}}{\mathrm{d}\mu_{U_i}}\left(\frac{1}{\mu_{U_i}}\right)\right)^2 \sigma_{U_i}^2 = \frac{1}{\mu_{U_i}^4}\sigma_{U_i}^2 \propto \frac{1}{\mu_{U_i}^4}.$$

Therefore, conditional on all other terms in Eq. (1), the variance of the model-predicted concentrations increases as a quartic of the true inverse wind speed. This is important, as it means that model predictions at low wind speeds, say less than $1\ \mathrm{m\ s^{-1}}$,

could be highly uncertain; we show a way of handling this uncertainty when we detail the Bayesian inversion model in Sect. 4.

### 3.3   Predicted concentrations for point and path measurements

The plume model given by Eq. (1) sets the $x$-axis as its centreline and the $\mathrm{CH_4}$ source at the origin. The predicted plume-model concentration at a physical location $(\tilde{x}_i, \tilde{y}_i, \tilde{z}_i)$ is thus found by first applying a spatial shift and time-dependent rotation (by wind direction) to $(\tilde{x}_i, \tilde{y}_i, \tilde{z}_i)$ in order to obtain $(x_i, y_i, z_i)$, which is then used to compute a model-predicted concentration

(conditional on $Q$, $U_i$, $H$, and $\boldsymbol{\theta}_{k_i}$). Conversion to ppm is done via the ideal gas law.

Let $C_i$ be a model-predicted concentration ($i = 1, 2, \ldots, N$). If $C_i$ corresponds to a point measurement, then one needs only to evaluate Eq. (1) at the transformed point-measurement location to obtain a predicted concentration. If $C_i$ corresponds to a path measurement, however, it represents an average of concentrations along the path. Denote the transformed end points of the straight-line path in the horizontal plane as $(x_{i,1}, y_{i,1})$ and $(x_{i,2}, y_{i,2})$, respectively. The line between the given points in

the horizontal plane can be parametrised by $\boldsymbol{\rho}_i(s) = (\rho_{x,i}(s), \rho_{y,i}(s))'$, where

$$\rho_{x,i}(s) = s x_{i,2} + (1-s) x_{i,1}, \quad s \in [0,1],$$
$$\rho_{y,i}(s) = s y_{i,2} + (1-s) y_{i,1}, \quad s \in [0,1],$$

so that

$$C_i = \frac{1}{T_i} \int_0^1 C(\rho_{x,i}(s), \rho_{y,i}(s), z_i, Q, U_i, H, \boldsymbol{\theta}_{k_i}) \left\| \frac{\mathrm{d}\boldsymbol{\rho}_i(s)}{\mathrm{d}s} \right\| \mathrm{d}s, \tag{2}$$

where $T_i \in \mathbb{R}^+$ is the path length and $\|\cdot\|$ is the standard Euclidean norm. In our case, $T_i = \left\| \frac{\mathrm{d}\boldsymbol{\rho}_i(s)}{\mathrm{d}s} \right\|$ is not a function of $s$, and so Eq. (2) simplifies to $C_i = \int_0^1 C(\rho_{x,i}(s), \rho_{y,i}(s), z_i, Q, U_i, H, \boldsymbol{\theta}_{k_i}) \mathrm{d}s$. This integral can be approximated numerically over a fine partitioning of $J$ segments $P = \{[s_0, s_1], [s_1, s_2], \ldots, [s_{J-1}, s_J]\}$, where $0 = s_0 < s_1 < \cdots < s_{J-1} < s_J = 1$. Then

$$C_i \approx \sum_{j=1}^{J} C(\rho_{x,i}(s_j^*), \rho_{y.i}(s_j^*), z_i, Q, U_i, H, \boldsymbol{\theta}_{k_i}) \Delta_s,$$

where $\Delta_s \equiv s_j - s_{j-1} = 1/J$ and $s_j^* = \frac{s_j + s_{j-1}}{2}$. In our experiments we set $J = 100$.

## 4   Bayesian atmospheric tomography

We are ultimately interested in obtaining a range of plausible values for the emission rate, $Q$, *a posteriori*, (i.e., after we have observed some data). In this section we present a hierarchical statistical model that relates $Q$ to the observed concentrations via the Gaussian plume model. Although $Q$ itself is univariate, the model contains several other unknown parameters that capture our uncertainty about the physical and the measurement processes; inferences on these parameters and $Q$ are made simultaneously. For ease of exposition we adopt the terminology of Berliner (1996) to describe the model, which we also summarise graphically in Fig. 3. The top layer in the hierarchy is the data model (the model for the observations, $\boldsymbol{Y}$, Sect. 4.1), the middle layer is the process model (the model for $Q$, Sect. 4.2), and the bottom layer is the parameter model (the unknown parameters not of direct interest, $\boldsymbol{\tau}$, $\omega_y, \omega_z$, Sect. 4.3). In Sect. 4.4 we outline the MCMC strategy we use to make inference with the model.

### 4.1   The data model

Let $\widetilde{\boldsymbol{Y}} \equiv (\widetilde{Y}_1, \widetilde{Y}_2, \ldots, \widetilde{Y}_N)'$ denote the measured concentrations averaged over five-minute intervals. We model each of these averaged measurements as $\widetilde{Y}_i = C_i + X_i + \varepsilon_i$, where $C_i$ is the $i$th Gaussian plume-predicted concentration, $X_i$ is the sum of the $i$th CH$_4$ background concentration and instrument-specific bias, and $\varepsilon_i$ denotes the random error associated with the $i$th observed CH$_4$ concentration. The background concentration and bias can be explicitly modelled and predicted (Ganesan et al., 2015). Here, as in Zammit-Mangion et al. (2015), we estimate $X_i$ as the 5th percentile of all the measurements from

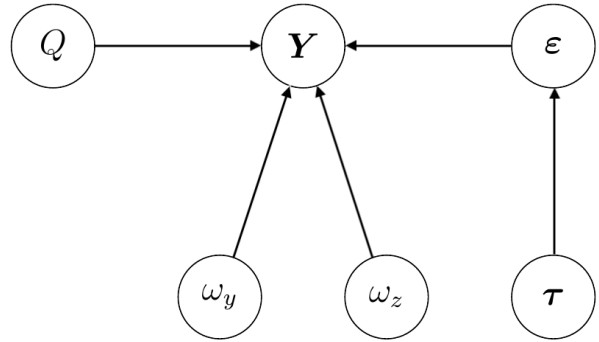

**Figure 3.** Directed acyclic graph showing the conditional dependence relationships between the data (enhancements) $\boldsymbol{Y}$ and the error components $\boldsymbol{\varepsilon}$ (Sect. 4.1), the emission rate $Q$ (Sect. 4.2), and the unknown parameters $\boldsymbol{\tau}, \omega_y$, and $\omega_z$ (Sect. 4.3).

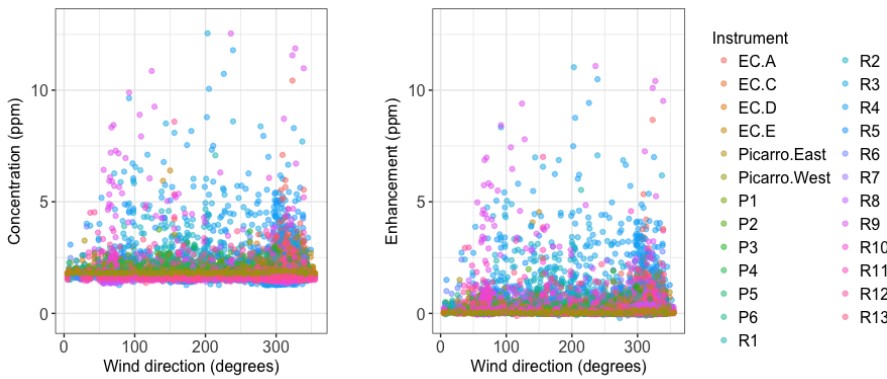

**Figure 4.** Left: Raw averaged concentrations, plotted by instrument and against wind direction. Right: Enhancements obtained by subtracting off the background and instrument-specific bias.

the instrument associated with the $i$th measurement. Figure 4 compares the raw averaged concentrations to those corrected for background and instrument-specific bias, which we term *enhancements*, when plotted against wind direction (in degrees East of North).

Now, let $\boldsymbol{Y} \equiv (Y_1, Y_2, \ldots, Y_N)'$ denote the enhancements. It is straightforward to verify that

$$
\begin{aligned}
Y_i &= \widetilde{Y}_i - X_i \\
&= C_i + \varepsilon_i, \quad i = 1, \ldots, N.
\end{aligned}
$$

Therefore, $Y_i$ is made up of two main components of variability: the Gaussian plume-predicted concentration and a random error term. We assume that the $\varepsilon_i, i = 1, \ldots, N$, are Gaussian and independent, but that they are not identically distributed. Specifically, $\varepsilon_i$ contains two components of variation, one pertaining to the error characteristics of the instrument, and one to

the stability class with which we have categorised the measurement. Recall also from Sect. 3.2 that we model the variance of the predicted concentrations to be proportional to a quartic of the true mean inverse wind speed for $U_i < 1$ m s$^{-1}$.

First, we capture instrument-specific measurement error characteristics and stability-condition-specific variation by introducing an auxiliary variable $m_i$ ($m_i = 1, 2, \ldots, M$), where $M$ is the total number of unique combinations of stability class and instrument type, and consider $M$ different precision (i.e., inverse variance) parameters $\{\tau_{m_i}\}$ that need to be estimated, one for each combination. Second, we take the influence of low wind speeds into account by assuming that the precision of $\varepsilon_i$ is $\tau_{m_i}$ multiplied by $\hat{U}_i$, where, for $U_i > 0$,

$$
\hat{U}_i = \begin{cases} U_i^4 & 0 < U_i < 1, \\ 1 & U_i \geq 1, \end{cases} \tag{3}
$$

which encapsulates our prior belief that observed model-measurement mismatch variability at low wind speeds (in this case under 1 m s$^{-1}$) are dominated by the low wind speed.

Putting these two components together, we have that, conditional on the instrument type and stability class encoded in $m_i$,

$$
\varepsilon_i \mid m_i \sim \text{Gau}(0, 1/(\hat{U}_i \tau_{m_i})), \quad i = 1, \ldots, N.
$$

We detail the prior distribution for $\tau_{m_i}$ in Sect. 4.3.1.

## 4.2 The process model

The process of interest in this application is the emission rate, $Q$, which we assume is constant. Since in this application $Q \geq 0$, we model it using a half-normal prior distribution (a Gaussian distribution with mean zero truncated from below at zero),

$$
p(Q) = \begin{cases} \frac{\sqrt{2}}{\sigma_Q \sqrt{\pi}} \exp\left(-\frac{Q^2}{2\sigma_Q^2}\right), & Q \in [0, \infty) \\ 0 & \text{otherwise,} \end{cases} \tag{4}
$$

with a standard deviation parameter, $\sigma_Q$, which is known and fixed. In our case we fixed $\sigma_Q$ to 1.5 g s$^{-1}$ (90 g min$^{-1}$) which results in a relatively uninformative prior distribution.

While addressing nonnegativity, half-normal priors do not contain a point mass at zero, and thus do not encode a prior belief that there is a possibility of having exactly a zero emission rate. As a consequence, a posterior estimate or even a credible interval that includes zero is not possible. A spike-and-slab distribution (Mitchell and Beauchamp, 1998) consisting of a diffuse uniform distribution with a point-mass at zero, could be alternatively used at the cost of a slightly more complex model.

## 4.3 The parameter model

Our parameter model is divided into two parts, one pertaining to the precision parameters $\{\tau_{m_i}\}$ in the random-error component in the data model, and the other to the standard-deviations in the Gaussian-plume dispersion models which, as shown in Sect. 3.1, are also uncertain.

### 4.3.1 The precision parameters

For conjugacy with the Gaussian likelihood, we model each $\tau_{m_i}$ using a Gamma prior distribution, with shape parameter $\alpha$, and rate parameter $\beta$:

$$p(\tau_{m_i}) = \frac{\beta^\alpha}{\Gamma(\alpha)} \tau_{m_i}^{\alpha-1} e^{-\beta \tau_{m_i}}, \quad i = 1, \dots, N.$$

In our application we set $\alpha = 1.058$ and $\beta = 0.621$. These values were chosen through quantile-matching, such that the 1st and 99th percentiles of the distribution of $1/\sqrt{\tau_{m_i}}$ are approximately 0.35 ppm and 6.5 ppm, respectively (giving a mode close to 0.7 ppm). Values for these percentiles were selected based on prior exploratory data analysis of the measurements that were taken upwind of the source.

### 4.3.2 The Gaussian plume model parameters

From separate studies into the reliability of the model values for $\sigma_{y_i,k_i}$ and $\sigma_{z_i,k_i}$, briefly discussed in Sect. 3.1, we concluded that these parameters could indeed be off by factors of two or more and that, if they are off, they are so by similar amounts for each stability class. These factors correspond to vertical shifts of the Pasquill stability curves when plotted on a log-log scale (e.g., Wark et al., 1998, Chapter 4). We thus replaced $\sigma_{y_i,k_i}$ and $\sigma_{z_i,k_i}$ in Eq. (1) with $\tilde{\sigma}_{y_i,k_i}$ and $\tilde{\sigma}_{z_i,k_i}$, respectively, where

$$\tilde{\sigma}_{y_i,k_i} \equiv \omega_y \sigma_{y_i,k_i} \qquad \text{and} \qquad \tilde{\sigma}_{z_i,k_i} \equiv \omega_z \sigma_{z_i,k_i},$$

and $\omega_y, \omega_z \in \mathbb{R}^+$ are scaling parameters for $\sigma_{y_i,k_i}$ and $\sigma_{z_i,k_i}$, respectively (Borysiewicz et al., 2012).

We use Gamma prior distributions for $\omega_y$ and $\omega_z$. In our application we set the shape parameters equal to 1.6084, and the rate parameters equal to 0.7361. These parameters give approximate 1st and 99th percentiles of 0.1 and 8, respectively, and a mode close to 1 (representative of no scalar influence on $\sigma_{y_i,k_i}$ or $\sigma_{z_i,k_i}$). This reflects our prior belief that the standard deviations could be up to an order of magnitude off from those derived using classical Pasquill stability-class theory.

### 4.4 Bayesian inference

Let $\boldsymbol{Y} \equiv (Y_1, Y_2, \dots, Y_N)'$ denote the $N$ observed enhancements. Similarly, define $\boldsymbol{U} \equiv (U_1, U_2, \dots, U_N)'$ and $\boldsymbol{\Theta} \equiv (\boldsymbol{\theta}_{k_1}, \boldsymbol{\theta}_{k_2}, \dots, \boldsymbol{\theta}_{k_N})'$. Further, let $\boldsymbol{\tau} \equiv (\tau_1, \tau_2, \dots, \tau_M)'$ be the $M$ parameters associated with each combination of instrument type and stability class. The posterior distribution of the emission rate $Q$ is then given by

$$p(Q \mid \boldsymbol{Y}, \boldsymbol{U}, H, \boldsymbol{\Theta}) \propto \int_0^\infty \int_0^\infty \int_{\mathbb{R}^{M+}} p(\boldsymbol{Y}, \boldsymbol{\tau}, \omega_y, \omega_z \mid Q, \boldsymbol{U}, H, \boldsymbol{\Theta}) p(Q) \, d\boldsymbol{\tau} \, d\omega_y \, d\omega_z$$

$$= p(Q) \int_0^\infty \int_0^\infty \int_{\mathbb{R}^{M+}} p(\boldsymbol{Y} \mid Q, \boldsymbol{\tau}, \omega_y, \omega_z, \boldsymbol{U}, H, \boldsymbol{\Theta}) p(\boldsymbol{\tau}) p(\omega_y) p(\omega_z) \, d\boldsymbol{\tau} \, d\omega_y \, d\omega_z,$$

where $p(Q)$ is given by Eq. (4) and $p(\boldsymbol{Y} \mid Q, \boldsymbol{\tau}, \omega_y, \omega_z, \boldsymbol{U}, H, \boldsymbol{\Theta})$ is the likelihood, which is Gaussian.

Computation of the posterior distribution $p(Q \mid \boldsymbol{Y}, \boldsymbol{U}, H, \boldsymbol{\Theta})$ involves a high-dimensional integral that is analytically intractable. We therefore use MCMC, specifically a Gibbs sampler, to obtain samples from the posterior distributions of $Q$, $\boldsymbol{\tau}$, $\omega_y$, and $\omega_z$ (see Gelman et al., 2013, for a comprehensive introduction to MCMC). The Gibbs sampler samples each parameter one at a time from their respective full conditional distributions, where conditioning is done using the most recent samples of all other parameters.

In the case of $\boldsymbol{\tau}$, use of Gamma prior distributions leads to full conditional distributions that are also Gamma. Hence, sampling $\boldsymbol{\tau}$ is straightforward. However, the prior distributions on the other parameters are not conjugate priors, and hence the full conditional distributions for each of these are not available in closed form. We therefore use standard Metropolis-within-Gibbs to sample from these conditional distributions, with Gaussian proposals and adaptive scaling during the early stages of the MCMC algorithm. Specifically, for each parameter, the standard deviation of the proposal was increased or decreased as appropriate whenever the acceptance rate fell below 10% or exceeded 80%.

## 5    Results and Discussion

### 5.1    Observing system simulation experiment

In this section we discuss results from applying our model to simulated data in an observing system simulation experiment (OSSE). To mimic the conditions in the real experiment, we simulated enhancements using the actual Boreal and EC instrument locations, meteorological observations from the Ginninderra data, and realistic variances for the random-error components. We considered the two release-rate periods separately, using a 6 g min$^{-1}$ emission rate in the first, and a 12 g min$^{-1}$ emission rate in the second. As in the real experiment, the first Boreal laser/reflector setup (seven paths) was used in the first release-rate period, while the second setup (six paths) was used in the second release-rate period; the EC tower locations were kept constant for both periods. We set the precisions $\tau_{m_i} = 4$ for $m_i = 1, \ldots, M$, and the scaling factors $\omega_y = \omega_z = 2$ to assess the algorithm's ability to calibrate the plume on-line. Following data simulation, we used MCMC to generate 60000 samples, left out 20000 of these as burn-in, and used a thinning factor of 10. Adaptation of the Metropolis samplers was only done during burn-in. Convergence was assessed through visual inspection of the MCMC trace plots.

We made inference on $Q$, as well as all other parameters in the model, for the Boreal- and EC-simulated data and the two emission rate settings. Table 2 shows the posterior median emission rates, the 95% posterior credible intervals for the emission rate, as well as the intervals for the plume standard-deviation scaling parameters $\omega_y$, and $\omega_z$. In all cases, we see that the true (simulated) emission rate is captured within our posterior credible intervals, and that the median estimates are very close to the true values. Interestingly, we see that while the plume-scaling coefficients have been accurately recovered in most cases, the posterior uncertainty over $\omega_y$ for the Boreals is very wide. This suggests that $\omega_y$ might not be identifiable for path measurements, possibly because the averaging effect of the line integral renders the measured concentration insensitive to a specific plume width in the horizontal direction.

**Table 2.** Posterior median emission rates in g min$^{-1}$, and the posterior 95% credible intervals of the emission rate in g min$^{-1}$, $\omega_y$, and $\omega_z$ from the OSSE. Results shown are from simulated data corresponding to the Boreals (B) and EC towers (E) when the emission rate is 6 g min$^{-1}$ (E1 and B1) and 12 g min$^{-1}$ (E2 and B2).

|    | Median $Q$ | $Q$ | $\omega_y$ | $\omega_z$ |
|----|----|----|----|----|
| B1 | 6.0718 | $(5.7847, 6.3511)$ | $(0.17978, 7.1675)$ | $(1.8871, 2.1378)$ |
| E1 | 6.0369 | $(5.5695, 6.5300)$ | $(1.7510, 2.1051)$ | $(1.7382, 2.2167)$ |
| B2 | 12.122 | $(11.820, 12.409)$ | $(0.17451, 6.7320)$ | $(1.9066, 2.0295)$ |
| E2 | 11.756 | $(10.884, 12.761)$ | $(1.7785, 2.0533)$ | $(1.7810, 2.2151)$ |

## 5.2 Application to the Ginninderra data set

In this section we discuss results from applying our model to enhancements from the compiled Ginninderra data. We considered several settings. In the first setting, we estimated the emission rate separately for each of the four instrument types, and for each release-rate period (5.8 g min$^{-1}$ and 5.0 g min$^{-1}$) when the source was active. In addition, for each release-rate period we estimated the emission rate for all the instruments combined, yielding a total of ten inversion results. In the second setting we estimated the emission rate for the same ten cases, but for periods when the source was switched off. In the third setting we again considered the same ten cases, but using only measurements that were taken when upwind of the source. These three settings serve to demonstrate how our inferences adapt to the various settings one might encounter in the field. In particular, online plume calibration is almost impossible in the latter two settings, and we expect this to result in large posterior uncertainties on the scaling coefficients, and also the emission rate in the third setting. In the second setting downwind measurements are present. Therefore, while online plume calibration is again almost impossible since there is no active source, the absence of a source ($Q = 0$) should be reflected in our posterior inferences (recall, however, that use of a half-normal prior distribution precludes the possibility of a zero emission rate being estimated; see Sect. 4.2).

As in the OSSE, we generated 60000 MCMC samples, left out 20000 of these as burn-in, and used a thinning factor of 10. In line with what we observed in the OSSE, our initial results showed that, more often than not, $\omega_y$ is not identifiable (leading to wide posterior distributions and poor MCMC mixing) when attempting to estimate the emission rate with the source switched on with *path* measurements. We therefore chose to fix $\omega_y = 1$ (but not $\omega_z$) for path measurements, and this choice is reflected in all the results discussed below.

The left panel of Fig. 5 summarises our results for $Q$ in the first setting (both upwind and downwind measurements with the source switched on); full results are given in the first ten rows of Table A1. While our posterior inferences are reflective of the true underlying emission rate, unlike in the OSSE we see that with the real data the true values were not always captured within our 95% posterior credible intervals. This suggests that there are other important factors at play (e.g., with the meteorological data such as ambient temperature or wind direction, that we assume are fixed and known) that are not (or not fully) accounted for in our model. A close inspection of the residuals at EC.A revealed mild deviations from our Gaussianity

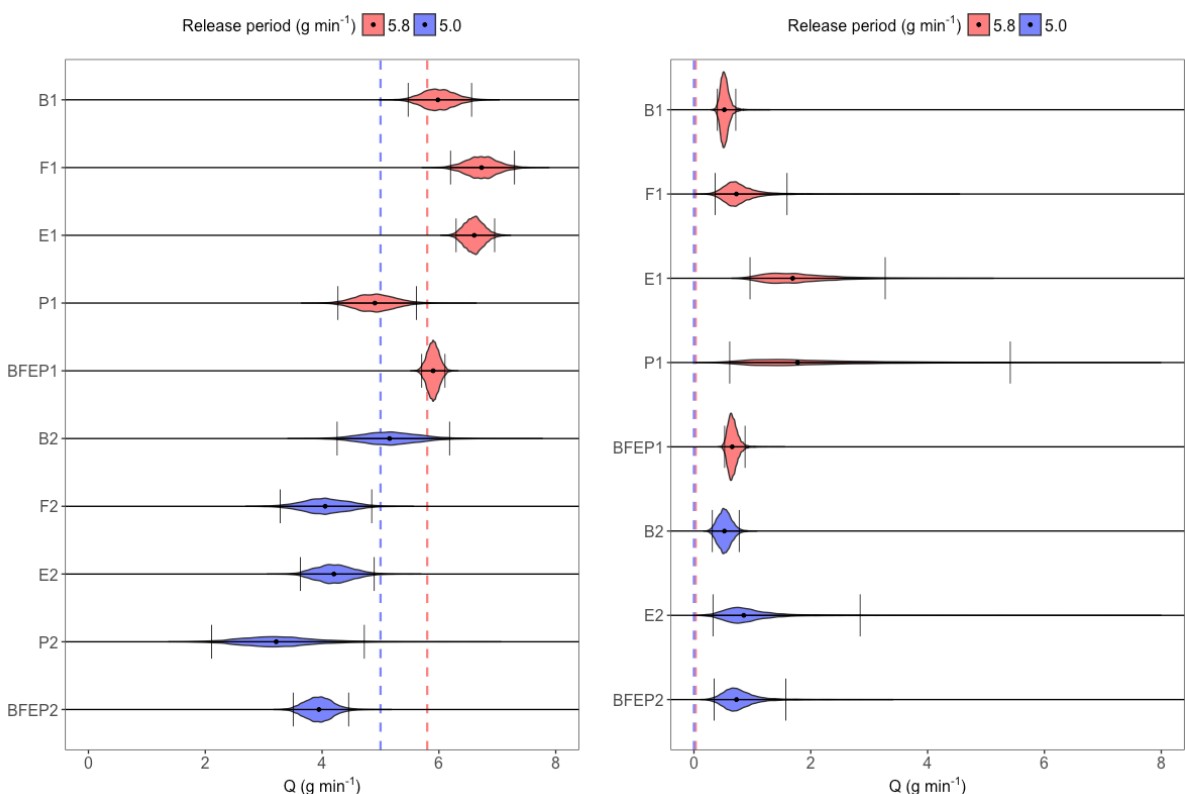

**Figure 5.** Left: posterior empirical distributions of the emission rate $Q$ in g min$^{-1}$, for the Boreals (B), FTIRs (F), EC towers (E), Picarro analysers (P), and the ensemble of all instruments (BFEP), for each release-rate period (1 and 2) during the Ginninderra experiment. The 5.8 g min$^{-1}$ release-rate period is shown in red (B1, F1, E1, P1, and BFEP1), while the 5.0 g min$^{-1}$ release-rate period is shown in blue (B2, F2, E2, P2, and BFEP2). The vertical dashed lines denote the respective true emission rates, the black dots denote the median estimates, and the black vertical bars denote the upper and lower limits of the 95% posterior credible intervals. Right: same as the left panel but showing results obtained using measurements taken when the methane point source was inactive. We can recover a reasonable range of estimates for the emission rate, with no 95% posterior credible interval being far from the true emission rate. Further, we see that the posterior emission rate credible intervals move towards zero when the source is inactive, as desired.

assumption, while posterior predictive distributions on left-out EC tower data in a re-analysis revealed coverage probabilities (specifically, empirical probabilities computed from the quantity of validation data falling into the 68% and 95% prediction intervals, respectively) that are slightly too large. Nevertheless, our worst-case scenario, obtained with the combination of all instruments in the 5.0 g min$^{-1}$ release-rate period, had an interval limit which was only 0.55 g min$^{-1}$ (approximately 11%) off from the true value, while all posterior medians were within 36% of the true value (within 22% if one ignored results from the Picarro analysers during the 5.0 g min$^{-1}$ release-rate period). This is encouraging because a single, common inference method was used to obtain the inferences from data at a common temporal resolution – no manual instrument-specific tuning

was carried out. The approach thus seems relatively robust to instrument type; in Sect. 6 we show this is no longer the case once certain components in our model are assumed fixed and known.

The first ten rows of Table A1 also show the 95% posterior credible intervals for $\omega_y$ and $\omega_z$. None of the obtained credible intervals for $\omega_y$ contain 1, and the results corroborate the conclusion from our exploratory data analysis in Sect. 3.1 that a

plausible value for $\omega_y$ is about 2 or 3. This result lends credence to our ability to calibrate the Pasquill stability-class curves corresponding to $\sigma_{y_i,k_i}$ while estimating the emission rate with point measurements. There was less agreement on $\omega_z$ in the inversions, suggesting that something more complex than a simple scaling is required (or that the model used for $\sigma_{z_i,k_i}$ is, in this case, inappropriate) for calibrating the Pasquill stability-class curves corresponding to $\sigma_{z_i,k_i}$. Nonetheless, in Sect. 6 we show that our emission-rate estimates from point measurements were relatively less sensitive to the assumption $\omega_z = 1$ than to

the assumption $\omega_y = 1$.

The right panel in Fig. 5 summarises our results for $Q$ in the second setting (both upwind and downwind measurements with the source switched off), while full results are given in the second set of ten rows in Table A1. Recall from Sect. 4.2 that due to the choice of prior over $Q$ (a half-normal distribution), it is not possible for the 95% credible interval to include zero. Clearly, however, the intervals for $Q$ are close to zero and are suggestive of a small emission rate. As expected, the plume

standard-deviation scaling parameters are not well-constrained in this setting when the source is off: Narrow credible intervals on the emission rate here are only possible when the measurement is largely insensitive to the plume shape. This is indeed the case for the Boreal paths, some of which pass very close to the source. With other instrument configurations, uncertainty in the plume scalings dominates. In some cases (FTIRs and Picarro analysers in the 5.0 g min$^{-1}$ release-rate period) our MCMC algorithm did not converge after the 60000 samples; these results are thus omitted from Fig. 5 and Table A1.

The bottom ten rows in Table A1 give full results in the third setting (upwind measurements only with the source switched on). In this setting the 95% posterior credible intervals produced for the emission rates are very wide (most with a range of over 100 g min$^{-1}$), as are those produced for $\omega_y$ and $\omega_z$: Our posterior distributions are largely uninformative. This was expected since upwind measurements contain no information on both the emission rate *and* the plume model parameters. These results from upwind measurements serve as verification, and confirm that we are indeed relying on useful information from downwind

measurements when making inference on the emission rate and other parameters that appear within our model.

## 6   Sensitivity of results to model components

As detailed throughout Sect. 4, the Bayesian model we employ contains many parameters that are updated using MCMC. A natural question to ask is whether all these parameters do need to be updated, and what the effects on the emission rate inferences are when instead some of these are assumed fixed and known. Specifically, we are interested in seeing what happens

when: (i) considering only one single precision parameter $\tau$ for all of the data regardless of stability class and/or instrument group; (ii) considering one $\tau_{m_i}$ per instrument group only; (iii) not accounting for plume-model variability in low wind speeds (i.e., setting $\hat{U} = 1$); (iv) not updating $\omega_y$ when using point measurements; (v) not updating $\omega_z$; and (vi) not updating both $\omega_y$

and $\omega_z$ when using point measurements. The 95% credible intervals for $Q$ in g min$^{-1}$ for all these settings and for each of the 10 groupings considered in Sect. 5 are given in Table A2.

Grouping the precision parameters $\{\tau_{m_i}\}$ by instrument only (instead of by instrument *and* stability class) had a slightly negative impact on the emission-rate estimates obtained during the second release-rate period, but less so during the first release-rate period. Assuming (and fixing) $\omega_z = 1$ for both the point and path measurements also did not have a serious impact on the emission-rate estimates. Note that this does not mean that these components are not relevant in the general model – for example, from our estimates of $\omega_z$ in Table A1 we see $\omega_z = 1$ would be a plausible choice for this experiment if one opted to fix $\omega_z$ (while $\omega_y = 1$ would not be).

On the other hand several components in our model appear to be crucial to obtaining reasonable emission-rate estimates. Using a single precision parameter to capture all observed variability due to measurement error and the stability-class categorisation clearly had a negative impact on our emission-rate estimates. Similarly, assuming the variability of the measurements is independent of wind speed when doing inversion resulted in 95% posterior credible intervals on the emission rate that are considerably shifted in the negative direction. A similar observation was made by Feitz et al. (2018, p. 207) when analysing data from the Boreal lasers. There, observations with wind speeds below 1.5 m s$^{-1}$ were removed to mitigate this effect.

The scaling factor $\omega_y$ is clearly also crucial for obtaining emission-rate estimates of practical significance for point measurements, with the ensuing emission-rate estimates often being off by nearly a factor of two when $\omega_y = 1$ is assumed. As expected, the width of the credible intervals on the emission rate decreased substantially when $\omega_y = \omega_z = 1$ was assumed, indicating that $\omega_y$ and $\omega_z$ play a big role in quantifying uncertainty on the emission rate. Therefore, as noted in other studies discussed in Sect. 1, incorporating uncertainty in the transport model by treating parameters within the model itself as uncertain (note that this is different from adding another component of variability in the data model, as is often done) is likely to have a positive impact on emission-rate estimates and uncertainty quantification.

## 7 Conclusions

In this article we have proposed a fully Bayesian model for atmospheric tomography that takes into account uncertainty in the data measurement process, the physical processes, and parameters appearing in the transport model, when estimating the emission rate. We see that the model is robust to different instrument types and configurations, and provides useful inferences on the emission rate and the plume dispersion model used. When applied to the Ginninderra data using a variety of instruments in different release-rate periods, we obtain 95% posterior credible intervals on the emission rate that either encapsulate the true emission rate, or that have a limit which is no more than 11% from the true value.

The methods developed in this study are ideal for quantifying local-scale leaks from industrial facilities or from the subsurface (e.g., well heads, buried pipelines or gas leakage up geological fractures and faults) where a surface leak has been detected but needs to be quantified. It can be used where physical access to the source location is limited, e.g., gas bubbling from a creek or where measurement is hazardous. Depending on the circumstance, detection of leakage can take many different forms, from visible bubble detection, optical gas imaging, handheld sniffers, noise detection, helicopters equipped with

lasers, drones equipped with gas sensors, to monitoring die-off in vegetation using remote sensing techniques. Surface leakage typically expresses as small, concentrated hotspots if sourced from the subsurface (Feitz et al., 2014; Forde et al., 2019), for which the quantification approach outlined in this article is ideally suited. Equipment placement can be optimised around the leakage site (i.e., prevailing upwind/downwind) for optimal quantification.

In most applications the number of sources, nor the source location, is known. As such, the framework we construct should be seen as a foundational building block that needs to be extended appropriately for each specific application. For example, if the source location is not known, then source localisation can be incorporated into the Bayesian framework as discussed by Humphries et al. (2012). If there are multiple possible sites, and these locations are not known, then the framework needs to be further extended to incorporate multiple Gaussian plume models (one for each site), and joint localisation/inversion will be

required. While these extensions are straightforward both mathematically and computationally, in practice they are unlikely to be effective for detection of leakage over large spatial scales. Gas fields or geological storage sites can cover areas of tens to hundreds of square kilometres. Unless there is a high density of sensors ($\approx 100$ m scale, van Leeuwen et al., 2013; Jenkins et al., 2016), the sensitivity of detection will be poor (Wilson et al., 2014; Luhar et al., 2014). It is however relatively straightforward to effectively extend the methodology to when the emission is from an area, rather than a point source.

Our work is closely connected to other atmospheric tomography techniques, but with some small, significant, differences. Luhar et al. (2014) used a backward Lagrangian particle model to simulate the trajectories of methane and carbon dioxide backwards in time to localise the source and estimate the emission rates. Their approach yielded good quality estimates for the methane emission rates, but highly uncertain estimates for the carbon dioxide emission rates and source location parameters. Twenty-three runs of the Lagrangian model required approximately one hour of computing time, and therefore their framework

becomes problematic with thousands of observations as we have in our study. More pertinently, online calibration of the atmospheric-transport model would be virtually impossible without the construction and use of a surrogate model or emulator (e.g., Harvey et al., 2018). In the study of Humphries et al. (2012), carbon dioxide and nitrous oxide emission rates and source locations were estimated relatively well. We do not consider the localisation problem, but otherwise extend their method to handle various instrument types and a number of extra levels of uncertainty. The case in our sensitivity analysis in which we

fix $\omega_y = \omega_z = 1$ yields a model that is structurally very similar to that of Humphries et al. (2012); we see from our results that having this hard constraint is not a tenable assumption in practice. Our work also has close connections with that of Ars et al. (2017) where the Pasquill stability class for an observation is chosen from a subset of appropriate stability classes, based on the best fit of model predicted values to observed values. While this may help fit the Gaussian plume dispersion model to the data, it does not take into account the uncertainty arising from stability-class choice. Further, if all plume model standard deviations

are off by a factor of two or more, there is a distinct possibility that no stability class yields a good fit. Online calibration of these standard deviations is needed to account for lack-of-fit arising from the the inherently simple Gaussian plume model.

Our results provide interesting insights into the design and monitoring of sensor networks for detecting and quantifying methane emissions. For example, our sensitivity analysis in Sect. 6 showed that estimates using the two Picarro analysers were particularly sensitive to assumptions made on the model plume parameters. Moreover, when uncertainty on these parameters

was considered, the release-rate estimates from these instruments tended to be uncertain. This is despite the Picarro analysers

being among the more accurate and expensive instruments used in the study. Uncertainty in our experiment is, as is often the case, dominated by that in the transport model. Hence, the number of instruments used, the proximity of the instruments to the source, and their configuration around the source, appear to be more important design criteria than instrument accuracy when the inferential target is emission-rate quantification of a point source. In particular, having more (less expensive) instruments

5 set up to cover many more possible wind directions is better than having only one or two more expensive instruments with which to monitor emissions. If one is limited to using a small number of instruments, then those giving path measurements are preferrable to those giving point measurements, as the former will be able to 'capture' a larger range of wind directions. Our results also provide insight on the transport model used. For example, close inspection of our posterior inferences for $\tau$ indicated that across all instrument groups and for both release-rate periods, the model-data mismatch was much lower for the

10 more neutral stability classes C and D, than for the more stable/unstable classes A and F.

The fully Bayesian framework we adopt is adaptable to various scenarios. We envision, for example, that source localisation (e.g., Humphries et al., 2012; Hirst et al., 2013) could be done in tandem with plume-model calibration within an inversion framework, provided several instruments in suitable configurations (as in the Ginninderra experiment) are available. Future work will also investigate how uncertainty in other meteorological variables such as wind-direction, as well as the stability-

15 class categorisation adopted (possibly via $z_0$) could be incorporated within the model.

*Code and data availability.* Software code and data are available at https://github.com/Lcartwright94/BayesianAT.

# Appendix A: Full results

**Table A1.** Posterior median emission rate in g min$^{-1}$, and the posterior 95% credible intervals for the emission rate in g min$^{-1}$, $\omega_y$, and $\omega_z$, for the Boreals (B), FTIRs (F), EC towers (E), Picarro analysers (P), and an ensemble of all instruments (BFEP), for each release-rate period (5.8 g min$^{-1}$ (1), and 5.0 g min$^{-1}$ (2)) under various settings. Dashes correspond to parameters that were not updated via MCMC. Results for which MCMC did not converge are marked as NA.

| Setting | Group | Median $Q$ | $Q$ | $\omega_y$ | $\omega_z$ |
|---|---|---|---|---|---|
| | B1 | 5.9833 | $(5.4733, 6.5593)$ | — | $(3.2062, 4.2104)$ |
| | F1 | 6.7301 | $(6.1985, 7.2937)$ | — | $(1.4347, 1.8164)$ |
| | E1 | 6.6048 | $(6.2942, 6.9537)$ | $(2.4946, 2.7848)$ | $(1.0868, 1.1954)$ |
| | P1 | 4.9028 | $(4.2710, 5.6136)$ | $(2.6065, 3.6707)$ | $(0.41664, 0.64341)$ |
| Source on | BFEP1 | 5.9008 | $(5.7050, 6.1038)$ | $(2.3360, 2.5640)$ | $(1.1944, 1.2989)$ |
| (Upwinds & Downwinds) | B2 | 5.1552 | $(4.2571, 6.1820)$ | — | $(0.84608, 1.1288)$ |
| | F2 | 4.0525 | $(3.2838, 4.8497)$ | — | $(0.66723, 1.0944)$ |
| | E2 | 4.2017 | $(3.6297, 4.8923)$ | $(1.4899, 2.1671)$ | $(0.90941, 1.0981)$ |
| | P2 | 3.2135 | $(2.1071, 4.7236)$ | $(2.0250, 5.2798)$ | $(0.34677, 0.63648)$ |
| | BFEP2 | 3.9455 | $(3.5054, 4.4543)$ | $(1.7138, 2.5325)$ | $(0.97964, 1.1437)$ |
| | B1 | 0.52073 | $(0.40106, 0.71608)$ | — | $(1.3051, 5.0262)$ |
| | F1 | 0.72641 | $(0.36438, 1.5935)$ | — | $(1.2565, 9.0531)$ |
| | E1 | 1.6906 | $(0.95997, 3.2742)$ | $(10.768, 21.971)$ | $(3.1036, 11.826)$ |
| | P1 | 1.7798 | $(0.61237, 5.6367)$ | $(3.3985, 13.853)$ | $(0.31311, 7.3589)$ |
| Source off | BFEP1 | 0.65416 | $(0.52512, 0.87510)$ | $(7.0545, 12.789)$ | $(2.2381, 5.3166)$ |
| (Upwinds & Downwinds) | B2 | 0.52202 | $(0.31479, 0.77494)$ | — | $(0.84995, 1.5319)$ |
| | F2 | NA | NA | — | NA |
| | E2 | 0.85549 | $(0.32681, 3.3683)$ | $(2.3136, 11.371)$ | $(0.50337, 7.9746)$ |
| | P2 | NA | NA | NA | NA |
| | BFEP2 | 0.72846 | $(0.34557, 1.5735)$ | $(2.7823, 9.5185)$ | $(0.97971, 7.1704)$ |
| | B1 | 62.452 | $(2.7445, 206.22)$ | — | $(0.16883, 6.8461)$ |
| | F1 | 61.651 | $(3.2040, 207.05)$ | — | $(0.17249, 6.5361)$ |
| | E1 | 16.136 | $(7.5484, 41.030)$ | $(4.5921, 6.9288)$ | $(1.2708, 8.7488)$ |
| | P1 | 22.913 | $(2.0052, 168.70)$ | $(0.15931, 9.2038)$ | $(0.23560, 7.7829)$ |
| Source on | BFEP1 | 15.723 | $(7.1673, 39.188)$ | $(4.7485, 6.9568)$ | $(1.4798, 8.8789)$ |
| (Upwinds only) | B2 | 88.353 | $(5.6799, 244.48)$ | — | $(0.27891, 6.0868)$ |
| | F2 | 58.568 | $(2.7772, 192.74)$ | — | $(0.18217, 6.4708)$ |
| | E2 | 39.728 | $(3.2357, 180.33)$ | $(0.23683, 5.7448)$ | $(0.19650, 6.8680)$ |
| | P2 | 42.996 | $(1.9088, 185.75)$ | $(0.13023, 5.2626)$ | $(0.18403, 6.9436)$ |
| | BFEP2 | 37.909 | $(2.9071, 186.65)$ | $(0.22048, 5.4364)$ | $(0.28261, 7.1149)$ |

**Table A2.** Posterior 95% credible intervals for the emission rates in g min$^{-1}$ for the Boreals (B), FTIRs (F), EC towers (E) , Picarro analysers (P), and an ensemble of all instruments (BFEP), for each release-rate period (5.8 g min$^{-1}$ (1), and 5.0 g min$^{-1}$ (2)), and for various alterations to the model as detailed in Sect. 6. Dashes correspond to redundant case (e.g., $\omega_y = 1$ was assumed for all path measurements in the full model).

| Group | Full model | Assuming $\tau_{m_i} = \tau$ for $m_i = 1, \ldots, M$ | Assuming $\{\tau_{m_i}\}$ are only instrument-group dependent | Assuming $\hat{U} = 1$ |
|---|---|---|---|---|
| B1 | $(5.4733, 6.5593)$ | — | $(4.7238, 5.6727)$ | $(2.6092, 3.1975)$ |
| F1 | $(6.1985, 7.2937)$ | — | $(5.9526, 7.1190)$ | $(3.6482, 4.7116)$ |
| E1 | $(6.2942, 6.9537)$ | — | $(6.2062, 7.0047)$ | $(5.4894, 5.9759)$ |
| P1 | $(4.2710, 5.6136)$ | — | $(4.8748, 6.1139)$ | $(2.9868, 3.9053)$ |
| BFEP1 | $(5.7050, 6.1038)$ | $(4.7252, 5.2433)$ | $(5.8133, 6.2731)$ | $(3.4032, 3.6424)$ |
| B2 | $(4.2571, 6.1820)$ | — | $(4.0863, 6.4436)$ | $(2.5337, 3.5319)$ |
| F2 | $(3.2838, 4.8497)$ | — | $(2.7180, 4.2555)$ | $(1.4055, 2.1349)$ |
| E2 | $(3.6297, 4.8923)$ | — | $(3.2692, 9.4560)$ | $(3.1329, 4.1516)$ |
| P2 | $(2.1071, 4.7236)$ | — | $(1.6784, 4.7147)$ | $(1.8451, 3.0813)$ |
| BFEP2 | $(3.5054, 4.4543)$ | $(2.5283, 3.4837)$ | $(2.3224, 3.2790)$ | $(1.9421, 2.4779)$ |

| Group | Assuming $\omega_y = 1$ | Assuming $\omega_z = 1$ | Assuming $\omega_y = \omega_z = 1$ |
|---|---|---|---|
| B1 | — | $(4.0341, 4.7974)$ | — |
| F1 | — | $(5.4152, 6.3851)$ | — |
| E1 | $(3.3635, 3.7084)$ | $(6.8646, 7.5289)$ | $(3.6129, 3.8937)$ |
| P1 | $(2.0142, 2.5424)$ | $(5.8880, 7.5225)$ | $(2.6043, 3.4691)$ |
| BFEP1 | $(3.6176, 3.8644)$ | $(5.9888, 6.3946)$ | $(3.8251, 4.0726)$ |
| B2 | — | $(4.3543, 5.7021)$ | — |
| F2 | — | $(3.2608, 4.7770)$ | — |
| E2 | $(2.6442, 3.5321)$ | $(3.6982, 4.7588)$ | $(2.7116, 3.3605)$ |
| P2 | $(0.93638, 5.2326)$ | $(1.8757, 4.2556)$ | $(0.91678, 1.9052)$ |
| BFEP2 | $(2.4699, 3.0227)$ | $(3.6202, 4.5213)$ | $(2.5319, 3.0744)$ |

*Author contributions.* LC compiled the data with the help of all authors and ran all the analyses. LC and AZM conducted the research. AF conceptualised and supervised the study. LC, AZM, and AF wrote the manuscript. SB conducted an initial investigation using a simplified version of the proposed model. IS, FP, TC, KN, TN, MK, SZ, NW and NMD acquired the field data for the study. All authors discussed the results and commented on the manuscript.

*Competing interests.* The authors declare that they have no conflict of interest.

*Acknowledgements.* LC acknowledges support of the Australian Government Research Training Program Scholarship. AZM is supported by an Australian Research Council (ARC) Discovery Early Career Research Award (DECRA), DE180100203. LC, AZM, and AF would like to acknowledge APR.Intern for facilitating the first five months of this modelling study. NMD is supported by an ARC Future Fellowship, FT180100327. All authors thank Gareth Davies for reviewing an earlier version of this manuscript. The Ginninderra field site was supported by the Australian Government through the Carbon Capture and Storage – Implementation budget measure. The authors also acknowledge funding for the research provided by the Australian Government through the CRC program and support from the CO2CRC. The National Geosequestration Laboratory is thanked for making the two Picarro instruments available for the study. We would like to thank Phil Dunbar and his staff (CSIRO Plant Industry) for maintaining the site and Dale Hughes (CSIRO) for his assistance with maintenance of the CSIRO EC tower. The authors also wish to acknowledge the assistance of Field Engineering Services at Geoscience Australia. Geoscience Australia and the Western Sydney University team would like to acknowledge Charles Jenkins (CSIRO) for early discussions about the atmospheric tomography line technique and the Australian Mathematical Sciences Institute. The University of Wollongong wishes to acknowledge Joel Wilson, Maximilien Desservettaz, and Ruhi Humphries for their assistance in the site operations. AF and IS publish with the permission of the CEO, Geoscience Australia.

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
