# Peer review of "Bayesian atmospheric tomography for detection and quantification of methane emissions: Application to data from the 2015 Ginninderra release experiment"

_Atmospheric Measurement Techniques, 2019_

## Referee Comment (RC1) · Anonymous Referee #1 · 6 Jun 2019

Overall, I think this article is well thought-out, well written, and addresses an important problem in source quantification. I have just a handful of suggestions and ideas for the authors.

Broader ideas and suggestions:

- I think it can be difficult to figure out which Copernicus journal is the best fit for an article like this one. I have also seen several articles on inverse methods in Geoscientific Model Development (GMD), another Copernicus journal. I don't

think it would be worth switching this article over to GMD at this juncture, but I do think it would be helpful to get guidance from the Copernicus editors on which journal they would recommend for future articles on inverse modeling.

- I would be curious to see how the model results would change if you excluded observations collected when wind speeds are low. Many existing articles simply exlude observations that are collected when wind speeds are low, and the authors point out that this decision and the wind speed cutoff are subjective. I would argue that the choice of weighting function for the low wind speed observations is also somewhat subjective. I would be curious if these low wind speed observations contributes anything to the emissions estimate relative to the traditional approach of exlcluding those observations. I think it could be interesting to run a case with these observations excluded (if and only if it is not a lot of work to do).

- I think it could be useful to include a greater discussion of when and how this statistical model could be applied to other inverse problems. The Ginninderra Controlled Release experiment is a relatively simple and controlled emissions set up, and most real world source quantification problems are likely to be more complicated. For example, there may be multiple leaky natural gas wells in a study area instead of a single point source. I think it could be helpful to lay out for the reader what types of problems you think this model would be well-suited for.

Specific line-by-line suggestions:

- Abstract, line 3: Could you be more specific with the term "misspecifications"? What kind of misspecifications appear at each stage of the inversion processing chain? I think some people in the inverse modeling community may also feel offended by the phrasing of this statement, especially given the very broad scope of this statement. Instead of phrasing the abstract this way, it could be more effective to explain why certain inverse problems are so challenging.

- Abstract, line 6: The word "tomography" is not common in the atmospheric inverse modeling community. I think I know what you mean, but it could be worth clarifying somewhere in the manuscript.

- Abstract, line 9: Perhaps "inverse modeling" would be better than "inversion"?

- Page 2, line 9: Consider moving the clause at the beginning of the sentence to the end of the sentence. I think this change might make the sentence flow better.

- Page 2, line 29-31: This sentence has a relatively complex structure that makes it a bit difficult to read. You might consider simplifying this sentence or splitting it into two sentences for easier readability.

- Page 3, line 4-5: I don't quite follow the point being made in this sentence. I know that it is important to evaluate atmospheric transport as part of inverse modeling, but I am not sure how that task is naturally part of classical inverse theory.

- Page 5, line 9: What is a Pasquill stability class? I am not positive that your reader will know this term.

- Page 6, line 8: What is a Monin-Obukhov length?

- Figure 2: What message should readers take away from this figure? It could be useful to include a one-sentence takeaway message in the caption.

- Fig. 3: The directional errors are difficult to see in this figure. Consider making the arrow directions larger.

- Pg 11, line 2: Is a half-normal distribution the same as a truncated normal distribution?

- Figure 5: There is a lot going on in this figure. It could be helpful to the reader to state the main takeaway message of this figure somewhere within the caption.

---

## Referee Comment (RC2) · Anonymous Referee #2 · 19 Jun 2019

This study investigates the performance of an inverse modelling algorithm for estimating greenhouse gas emissions from a single point source using data from a controlled release experiment. Inverse modelling methods are widely used to quantify emissions addressing a large range of scales. The advantage of the small scale investigated here is that true emission can be known, allowing a direct evaluation of the inversion performance. Usually this is not possible and the performance can only be evaluated indirectly. The outcome is rather sobering, emphasizing the difficulty to obtain reliable emission estimates despite the favorable availability of data from different types

of instruments. An attempt is made to identify the most important factors limiting the performance. As explained in further detail below, it is unclear what we learn from this study that was not already known before. In part this is because the link is missing with earlier work, and how the performance that is achieved here compares with what was done in the past. It remains unclear also how well the optimized model is fitting the data and what would be needed to further improve the results. Further efforts in these directions will be needed to make this work publishable.

GENERAL COMMENTS

This study is following up on the study of Feitz et al (2018), in which not only the measurement techniques are described in detail but also different methods are used for emission quantification. That study is referenced, but it remains unclear how the method in this study relates to what was done before, and how the results compare. Besides the Ginninderra release experiment, other similar studies have been conducted in the past. To keep track of progress, and make sure the recommendations of those studies are taken into consideration it is important to make a closer connection to them and compare the performance that is achieved here.

This study arrives at the expected outcome that the performance of the inversion improves when the stability parameters of the Gaussian plume model are optimized. However, the finding that the performance of the OSSE is so much better than the results of the experiment using real data, despite using realistic settings in the OSSE, points to a significant remaining problem with the model. Given the simplicity of the Gaussian plume formulation this may not come as a surprise. Nevertheless, some further analysis of fit residuals is needed to find out what prevents the inversion from finding the right answer. Could it be as simple as the assumed wind direction or speed being wrong? I didn't find back an exact specification of the information that was used for that. How appropriate is the use of a Gaussian plume model in this experiment?

Different measurement techniques are compared, but there is very limited discussion

on the best technique for inferring emissions. What would be the recommendation for monitoring leaks? It would also be an option to use data that are not used in the inversion for evaluating the optimized concentrations.

SPECIFIC COMMENTS

Page 1: Introduction section: Here I was missing some information about the specific application of inverse modelling that is studied here, among the large range of applications discussed in literature. Special for this case is the small scale of the experiment and the known location of the emission source. Usually this is not the case, raising the question for which kind of application this would be relevant (you might argue that there are easier methods to monitor emissions when the source is known).

Page 6, line 22: '... serves to scale outputs vertically ...' In the end I understood that 'vertically' referred to the y-axis in figure 2. What is meant is that Q scales the concentration enhancements. Please rephrase to make this clearer.

Page 7, line 18: I'm missing an explanation of the logic here. Please clarify the reason for quantifying the statistics of 1/U.

Page 9, line 7: '... graphically in Fig. 3...." Here the dependent variables of the optimization are introduced, but not explained. Here the meaning of tau and the omega's should be explained explicitly, and that in addition to these variables Q is estimated from the data.

Page 10, line 2: '... and one to the stability class ...'. The model error contribution to $e\_i$ is not just the stability class.

Page 10, line 3: "(variance)" does not correspond to "(increases)". This is a good example why this grammar style is better avoided.

Page 10: What justifies a windspeed independent error for windspeeds > 1 m/s?

Page 11, line 7: Is a 'point mass at zero' not just a 'point emission of zero'. If so then

please change to avoid confusion.

Page 11, line 21: What is the relevance of the distribution of the inverse of the square root of the precision parameter? For a precision it would be straightforward to relate it to the numbers that are given in the same sentence. However, what is done here is more complicated for a reason that I don't see.

Page 11, line 7: 'While addressing . . . zero emission rate' Looking at equation 4, I don't see why it precludes zero as it is in the interval where the half-normal prior applies.

Page 14, figure 5: It is explained in the text why the method precludes zero as a solution to the inverse problem (see my earlier comment on that). However, I had expected the estimates to be much closer to zero when emissions are switched off. The likely reason is not the zero condition, but the accuracy at which the background concentration is accounted for. It makes me wonder why the background is not fitted as an additional unknown parameter.

Page 15, line 23: why are wind speed and direction assumed to be known?

TECHNICAL CORRECTIONS

Page 3, line 6: 'Houweling' instead of 'Houwelling'
* * *

---

## Author Comment (AC1) · 25 Jul 2019

Please see our responses to Reviewer 1 in Sect. 1 of the attached supplementary file.

Please also note the supplement to this comment:
https://www.atmos-meas-tech-discuss.net/amt-2019-124/amt-2019-124-AC1-supplement.pdf

---

## Author Comment (AC2) · 25 Jul 2019

**Authors' responses to the reviews of "Bayesian atmospheric tomography for detection and quantification of methane emissions: Application to data from the 2015 Ginninderra release experiment"**

(https://doi.org/10.5194/amt-2019-124)

**1 Reviewer 1**

**Overall, I think this article is well thought-out, well written, and addresses an important problem in source quantification. I have just a handful of suggestions and ideas for the authors.**

We thank the reviewer for taking the time to read and comment on the manuscript, and for suggesting improvements to the paper where needed.

**1.1 Broader ideas and suggestions**

**I think it can be difficult to figure out which Copernicus journal is the best fit for an article like this one. I have also seen several articles on inverse methods in Geoscientific Model Development (GMD), another Copernicus journal. I dont think it would be worth switching this article over to GMD at this juncture, but I do think it would be helpful to get guidance from the Copernicus editors on which journal they would recommend for future articles on inverse modeling.**

The ICDC10 Special Issue is an inter-journal issue which includes, amongst others, AMT. We believe the results in this manuscript are of relevance to the measurement-techniques community, since we consider the use of various instruments, in isolation and in combination, for detecting and quantifying methane emissions. We defer any decision on whether another journal in the issue is deemed a better fit to the AMT and Special Issue editors.

**I would be curious to see how the model results would change if you excluded observations collected when wind speeds are low. Many existing articles simply exlude observations that are collected when wind speeds are low, and the authors point out that this decision and the wind speed cutoff are subjective. I would argue that the choice of weighting function for the low wind speed observations is also somewhat subjective.**

The reviewer is correct in that both procedures are somewhat subjective. However, there are several advantages to downweighting the observations rather than excluding them. First, as we

show next, while low wind speed observations are somewhat unreliable, they still contain information of use and which could be important when optimising parameters within the Gaussian plume dispersion model and when doing flux inversion. Second, the way in which we downweight is motivated by transport-model considerations. In this sense we view downweighting as less subjective than excluding the observations entirely. We elaborate further in our reply to the next comment.

**I would be curious if these low wind speed observations contributes anything to the emissions estimate relative to the traditional approach of exlcluding those observations. I think it could be interesting to run a case with these observations excluded (if and only if it is not a lot of work to do).**

We have run the inversion using the EC tower data, during both release periods, and excluding all observations with wind speeds below 1 m s$^{-1}$ (which is also the value we use as a threshold in our downweighting scheme). The left panel of Fig. 1 in this response shows the original 95% posterior credible intervals obtained when downweighting observations with low wind speeds, and the right panel of Fig. 1 in this response shows the 95% posterior credible intervals when excluding the observations. There is only a small change between the results during the 5.8 g min$^{-1}$ release period. However, in the case of the 5.0 g min$^{-1}$ release period, there is a notable difference between the two posterior distributions. This is probably because there are far fewer observations during this release period than during the 5.8 g min$^{-1}$ release period. For the 5.8 g min$^{-1}$ release period, 2967 observations were included in the inversion which downweighted low wind speed observations, and 2460 observations were used when low wind speed observations were excluded. During the 5.0 g min$^{-1}$ release period, 475 observations were used when low wind speed observations were downweighted, compared to 339 observations when low wind speed observations were excluded. The smaller amount of information available in the 5.0 g min$^{-1}$ release period, along with so many unknown parameters to optimise, appears to have had a deleterious effect on the results.

We have also run the inversions for the two approaches with the threshold increased to 1.5 m s$^{-1}$. The results obtained using downweighting are shown in the left panel of Fig. 2 in this response. We see that the 95% posterior credible intervals for both release-rate periods have shifted by a small amount closer to the true release rate. The results for when data are discarded completely are shown in the right panel of Fig. 2 in this response. The 95% posterior credible interval is slightly wider than that of the downweighting method for the 5.8 g min$^{-1}$ release period. In both cases, the posterior distributions do not seem to be overly sensitive to the choice of threshold.

However, the Markov chains for the 5.0 g min$^{-1}$ release rate period and 1.5 m s$^{-1}$ threshold did not converge when $\omega_y$ and $\omega_z$ were treated as unknown and data were completely discarded, probably due to the lower amount of available observations. The results shown are therefore conditional on these parameters being fixed to the maximum a-posteriori estimates obtained when downweighting. This lack of convergence gives us more reason to believe that downweighting is preferable to discarding observations entirely.

We have now modified the first paragraph of Sect. 3.2 in the manuscript to indicate that (i) in separate studies we have seen that our posterior inferences are not overly sensitive to the choice of threshold used and that (ii) downweighting instead of discarding is particularly important when there are not many observations at high wind speed available for inference.

**I think it could be useful to include a greater discussion of when and how this statistical model could be applied to other inverse problems. The Ginninderra**

[Figure]

Figure 1: Left: posterior empirical distributions of the emission rate $Q$ in g min$^{-1}$, for the EC towers (E), for each release-rate period (1 and 2) during the Ginninderra experiment. Observations with wind speeds below 1 m s$^{-1}$ have been downweighted. The 5.8 g min$^{-1}$ release-rate period is shown in red (E1), while the 5.0 g min$^{-1}$ release-rate period is shown in blue (E2). The vertical dashed lines denote the respective true emission rates, the black dot represents the median estimate, and the vertical black bars represent the upper and lower limits of the 95% posterior credible interval. Right: same as left, however observations with wind speeds below 1 m s$^{-1}$ have been removed entirely.

[Figure]

Figure 2: Same as Fig. 1, however in the left panel observations with wind speeds below 1.5 m s$^{-1}$ have been downweighted, and in the right panel observations with wind speeds below 1.5 m s$^{-1}$ have been removed entirely.

**Controlled Release experiment is a relatively simple and controlled emissions set up, and most real world source quantification problems are likely to be more complicated. For example, there may be multiple leaky natural gas wells in a study area instead of a single point source. I think it could be helpful to lay out for the reader what types of problems you think this model would be well-suited for.**

The methods developed in this study are ideal for quantifying local-scale leaks from industrial facilities or from the subsurface (e.g., well heads, buried pipelines or gas leakage up geological fractures and faults) where a surface leak has been detected but needs to be quantified. It can be used where physical access to the source location is limited, e.g., gas bubbling from a creek or where measurement is hazardous. Depending on the circumstance, detection of leakage can take many different forms, from visible bubble detection, optical gas imaging, handheld sniffers, noise detection, helicopters equipped with lasers, drones equipped with gas sensors, to monitoring die-off in vegetation using remote sensing techniques. Surface leakage typically expresses as small, concentrated hotspots if sourced from the subsurface (Feitz et al., 2014; Forde et al., 2019), for which the quantification approach outlined in this article ideally suited. Equipment placement can be optimised around the leakage site (i.e., prevailing upwind/downwind) for optimal quantification.

In most applications the number of sources, nor the source location, is known. As such, the framework we construct should be seen as a foundational building block that needs to be extended appropriately for each specific application. For example, if the source location is not known, then source localisation can be incorporated into the Bayesian framework as discussed by Humphries et al. (2012). If there are multiple possible sites, and these locations are not known, then the framework needs to be further extended to incorporate multiple Gaussian plume models (one for each site), and joint localisation/inversion will be required. While these extensions are straightforward both mathematically and computationally, in practice they are unlikely to be effective for detection of leakage over large spatial scales. Gas fields or geological storage sites can cover areas of tens to hundreds of square kilometres. Unless there is a high density of sensors ($\sim 100$ m scale) (Jenkins et al., 2016; van Leeuwen et al., 2013), the sensitivity of detection will be poor (Luhar et al., 2014; Wilson et al., 2014). It is however relatively straightforward to effectively extend the methodology to when the emission is from an area, rather than a point source.

We now provide this discussion in Sect. 7 of the manuscript.

**1.2 Specific line-by-line suggestions**

**Abstract, line 3: Could you be more specific with the term "misspecifications"? What kind of misspecifications appear at each stage of the inversion processing chain? I think some people in the inverse modeling community may also feel offended by the phrasing of this statement, especially given the very broad scope of this statement. Instead of phrasing the abstract this way, it could be more effective to explain why certain inverse problems are so challenging.**

We did not intend to cause any offence: The term "model misspecification" has a technical meaning in statistical science, and refers to the fact that all statistical models (especially simple ones, like the one we are considering) are only approximations to the truth. For example, Rao (1971) states that model "[m]isspecification can arise either because of omission of a variable specified by the truth, [...] or because of inclusion of a variable not specified by the truth[.]" In addition to the statistical model being misspecified, it is also likely that the Gaussian plume

model is misspecified, in the sense that it does not (and can not) precisely capture the true dispersion of methane. To avoid ambiguity we have now used the word 'assumptions' instead of 'misspecifications' in the abstract.

**Abstract, line 6: The word tomography is not common in the atmospheric inverse modeling community. I think I know what you mean, but it could be worth clarifying somewhere in the manuscript.**

Atmospheric tomography was originally used by Humphries et al. (2012) due to its similarity to "tomographic imaging", where the aim is to detect and quantify the strength of sources in a medical context. We use this term to specify which problem of flux inversion (which often refers to the general problem of estimating fluxes over a region) we are concerned about in this manuscript. We have included the following sentence briefly describing the atmospheric tomography technique in the second last paragraph of Sect. 1 of the manuscript to clarify:

Atmospheric tomography, a term inspired from medical imaging, combines data from a collection of measurement sites with Bayesian inversion to detect and quantify emissions.

**Abstract, line 9: Perhaps inverse modeling would be better than inversion?**

Throughout the manuscript we use "inversion" synonymously with "inference", which we see as distinct from the "model" (or "inverse model"), which comprises the set of equations on which our inferences are based. We thus intentionally refrain from using the term "inverse model" in this context.

**Page 2, line 9: Consider moving the clause at the beginning of the sentence to the end of the sentence. I think this change might make the sentence flow better.**

This sentence has now been re-worded.

**Page 2, line 29-31: This sentence has a relatively complex structure that makes it a bit difficult to read. You might consider simplifying this sentence or splitting it into two sentences for easier readability.**

We now broke the sentence into two smaller sentences such that one key point is made in each sentence, as suggested. The first notes that online calibration of model parameters is not often done, while the second suggests a reason for this.

**Page 3, line 4-5: I dont quite follow the point being made in this sentence. I know that it is important to evaluate atmospheric transport as part of inverse modeling, but I am not sure how that task is naturally part of classical inverse theory.**

We apologise for the lack of clarity in this sentence. Evaluating the transport model is important for inverse modelling, but actually it is "calibrating the model from observations" that fits squarely within the Bayesian framework. The sentence has been rephrased to make this clearer.

**Page 5, line 9: What is a Pasquill stability class? I am not positive that your reader will know this term.**

The Pasquill stability classes are a collection of stability categories into which each observation can be placed. The categorisation of an observation is based on the values of the meteorological parameters at the time (for example, wind speed, wind direction, air temperature, air pressure), and how these values then affect the horizontal and vertical dispersion of the atmospheric particles. The more the particles are expected to spread in the horizontal and/or vertical

directions for a particular observation, the more unstable the Pasquill stability class associated with that observation. More details on the classes are given in Pasquill (1961), referenced in Sect. 3.1 of the manuscript. We have now added a note in Sect. 1, paragraph 3 that both the stability classes and the Monin-Obhukov length are described later in the manuscript. Further, in Sect. 3.1, we have now included a short description of the Pasquill stability class along with the reference to Pasquill (1961).

**Page 6, line 8: What is a Monin-Obukhov length?**

The Monin-Obukhov length is the theoretical height at which turbulence is produced by buoyancy and mechanical forces in equal amounts (Sienfeld and Pandis, 2006, Chapter 16). Hence, it is a continuous measure of stability. We have included a brief explanation of the $L$-value in the manuscript in Sect. 3.1.

**Figure 2: What message should readers take away from this figure? It could be useful to include a one-sentence takeaway message in the caption.**

The following sentence has been added highlighting the takeaway message:

Of the three, the best agreement between predicted and observed values occurs when $\sigma_{y_i,k_i}$ is scaled by 2.5.

**Fig. 3: The directional errors are difficult to see in this figure. Consider making the arrow directions larger.**

Thank you, the size of the arrows has now been increased.

**Pg 11, line 2: Is a half-normal distribution the same as a truncated normal distribution?**

The half-normal distribution is a special case of the truncated normal distribution. If a normal distribution with mean zero is truncated from below at zero, then it is a half-normal distribution. We have clarified this in Sect. 4.2 of the manuscript.

**Figure 5: There is a lot going on in this figure. It could be helpful to the reader to state the main takeaway message of this figure somewhere within the caption.**

Thank you, the following sentence has been added highlighting the takeaway message:

We can recover a reasonable range of estimates for the emission rate, with no 95% posterior credible interval being far from the true emission rate. Further, we see that the posterior emission rate credible intervals move towards zero when the source is inactive, as desired.

**2 Reviewer 2**

This study investigates the performance of an inverse modelling algorithm for estimating greenhouse gas emissions from a single point source using data from a controlled release experiment. Inverse modelling methods are widely used to quantify emissions addressing a large range of scales. The advantage of the small scale investigated here is that true emission can be known, allowing a direct evaluation of the inversion performance. Usually this is not possible and the performance can only be evaluated indirectly. The outcome is rather sobering, emphasizing the difficulty to obtain reliable emission estimates despite the favorable availability of data from different types of instruments. An attempt is made to identify the most important factors limiting the performance.

We thank the reviewer for taking the time to read and comment on the manuscript, and for suggesting improvements to the paper where needed.

As explained in further detail below, it is unclear what we learn from this study that was not already known before. In part this is because the link is missing with earlier work, and how the performance that is achieved here compares with what was done in the past. It remains unclear also how well the optimized model is fitting the data and what would be needed to further improve the results. Further efforts in these directions will be needed to make this work publishable.

We thank the reviewer for the review. There are several concerns raised in the introduction to the review, namely concerning

(i) Clarity on what we learn on the study;

(ii) How this compares to what is done now and in the past;

(iii) The quality of the fits; and

(iv) What could be done to improve the results.

Issues (i)–(iv) appear in the "General Comments" and "Specific Comments" below, and are addressed in our responses there.

**2.1 General comments**

This study is following up on the study of Feitz et al (2018), in which not only the measurement techniques are described in detail but also different methods are used for emission quantification. That study is referenced, but it remains unclear how the method in this study relates to what was done before, and how the results compare.

This comment concerns issue (ii) raised in the introduction to this review. This study builds upon the atmospheric tomography technique described in Section 2.4.2 of Feitz et al. (2018). The method has been refined in the current study through the "online calibration" of parameters within the atmospheric transport model. Further, in the previous study, atmospheric tomography was only used on the Boreal lasers, and the methodology was tailored to suit those specific instruments. The methodology we present accounts for model error and instrument-specific bias, and is applicable to both point- and path-measurements. It is relatively robust

to the type of instrument, with only the decision to include $\omega_y$ as an unknown parameter dependent on whether the observation is a point or path measurement.

We have now re-worded the second last paragraph in Sect. 1 of the manuscript to better detail the connection of this study with the previous study.

**Besides the Ginninderra release experiment, other similar studies have been conducted in the past. To keep track of progress, and make sure the recommendations of those studies are taken into consideration it is important to make a closer connection to them and compare the performance that is achieved here.**

This comment concerns issue (ii) raised in the introduction to this review. The reviewer is correct that other similar studies have been carried out, and we cite two of these in the manuscript: that of Humphries et al. (2012) and that of Luhar et al. (2014). A more comprehensive list of controlled release experiments and related studies has now been included in the second paragraph of Sect. 1. In addition, we have now included a paragraph detailing the connections of our work to those of Ars et al. (2017); Humphries et al. (2012); Luhar et al. (2014) in Sect. 7 of the manuscript:

Our work is closely connected to other atmospheric tomography techniques, but with some small, significant, differences. Luhar et al. (2014) used a backward Lagrangian particle model to simulate the trajectories of methane and carbon dioxide backwards in time to localise the source and estimate the emission rates. Their approach yielded good quality estimates for the methane emission rates, but highly uncertain estimates for the carbon dioxide emission rates and source location parameters. Twenty-three runs of the Lagrangian model required approximately one hour of computing time, and therefore their framework becomes problematic with thousands of observations as we have in our study. More pertinently, online calibration of the atmospheric-transport model would be virtually impossible without the construction and use of a surrogate model or emulator (e.g., Harvey et al., 2018). In the study of Humphries et al. (2012), carbon dioxide and nitrous oxide emission rates and source locations were estimated relatively well. We do not consider the localisation problem, but otherwise extend their method to handle various instrument types and a number of extra levels of uncertainty. The case in our sensitivity analysis in which we fix $\omega_y = \omega_z = 1$ yields a model that is structurally very similar to that of Humphries et al. (2012); we see from our results that having this hard constraint is not a tenable assumption in practice. Our work also has close connections with that of Ars et al. (2017) where the Pasquill stability class for an observation is chosen from a subset of appropriate stability classes, based on the best fit of model predicted values to observed values. While this may help fit the Gaussian plume dispersion model to the data, it does not take into account the uncertainty arising from stability-class choice. Further, if all plume model standard deviations are off by a factor of two or more, there is a distinct possibility that no stability class yields a good fit. Online calibration of these standard deviations is needed to account for lack-of-fit arising from the the inherently simple Gaussian plume model.

**This study arrives at the expected outcome that the performance of the inversion improves when the stability parameters of the Gaussian plume model are optimized. However, the finding that the performance of the OSSE is so much better than the results of the experiment using real data, despite using realistic settings in the OSSE, points to a significant remaining problem with the model. Given the simplicity of the Gaussian plume formulation this may not come as a surprise. Nevertheless, some further analysis of fit residuals is needed to find out what prevents the inversion from finding the right answer. Could it be as simple as the assumed**

[Figure]

[Figure]

Figure 3: Q-Q plots of the residual (observed enhancements minus the Gaussian plume dispersion model predicted values). All data is from EC.A, without wind speeds below 1 m s$^{-1}$, as these are not assumed to have constant variance. The left panel are observations from stability class A, and the right panel are observations from stability class F. Deviations away from the solid line indicate quantile deviations from the Gaussian distribution of same mean and variance.

**wind direction or speed being wrong? I didnt find back an exact specification of the information that was used for that.**

This comment concerns issue (iii) raised in the introduction to this review. It is important to note that the OSSE solely aims to verify that, under ideal conditions and no model misspecification, the inferential algorithm performs as it should. Indeed, one should not expect that results from real data are as good as those obtained in an OSSE. For example, it is likely that the Gaussian plume model predicted output is not valid for every observation we consider. The most we can hope for is a framework that is relatively robust to model misspecification, and this seems to be the case since our posterior inferences corroborate quite well with the truth.

The reviewer suggests that the wind data might be wrong. In the Ginninderra experiment weather information was recorded on site at EC.A, which was equipped with a number of meteorological instruments, as detailed toward the end of the first paragraph in Sect. 2 of our manuscript. In a separate study we had assumed that the wind speeds were uncertain. This dramatically increased the computational time required for MCMC, but had a negligible effect on the results. However, we have reason to believe that the wind speed and wind direction data are of good quality; we give more details in our response to the reviewer's final Specific Comment in this document.

Figure 3 in this response shows a Q-Q plot (a plot of theoretical quantiles against the observed quantiles) of the residuals obtained during the 5.8 g min$^{-1}$ release-rate period, for EC.A and stability classes A and F, and excluding observations with wind speeds below 1 m s$^{-1}$ (this is necessary because the variance is modelled to be wind-speed dependent below this threshold). The residuals suggest that there is a mild deviation away from our Gaussianity assumption. However, with real data, such mild deviations are to be expected, and the deviations do not appear large enough to be of concern. We have now noted that we have looked at the Q-Q plots and observed mild deviations from Gaussianity in the residuals in Sect. 5.2 of the manuscript (third paragraph).

**How appropriate is the use of a Gaussian plume model in this experiment?**

As mentioned in Sect. 1, paragraph 4 of our manuscript, the Gaussian plume model is known to

[Figure]

Figure 4: Radial histogram showing the direction *from* which the wind is blowing during the 5.8 g min$^{-1}$ release-rate period.

work well over short distances/small areas, such as the area used in the Ginninderra experiment (e.g., see Riddick et al., 2017). Moreover, the methodology we propose is aimed at applications where near-real time leak detection is important. In order to do this, computationally fast methods are necessary; an attraction of the Gaussian plume model is that it is very quick to evaluate and calibrate.

**Different measurement techniques are compared, but there is very limited discussion on the best technique for inferring emissions. What would be the recommendation for monitoring leaks?**

This comment concerns issue (i) raised in the introduction to this review. We see in our inversion results that the wind direction plays a large role in our ability to obtain reasonable estimates for the emission rate. For example, Fig. 4 in this response shows that the dominant wind directions during the 5.8 g min$^{-1}$ release-rate period range from approximately 300 to 340 degrees east of north. This suggests that very little of the released methane would have blown over Picarro.West, for example, while much of it would have blown over Picarro.East. Indeed, when the inversion is run on each of these instruments separately during this release-rate period, we see that a reasonable range of emission rate estimates is produced using Picarro.East. On the other hand, running the inversion using only Picarro.West returns the prior distribution set on $Q$, meaning these observations are not at all informative of the emission rate.

Therefore having more (less expensive) instruments set up to cover many more possible wind directions appears to serve better than having only one or two more expensive instruments. That is, unless the wind direction is favourable during a good portion of the experimental time frame, better coverage of the experimental field using less expensive/accurate instruments is likely to outperform a single, very expensive/accurate instrument when it comes to detecting and quantifying the emission rate from a point source. If one is limited to using a small number of instruments, then path measurements are more suitable than point measurements, since they are able to 'capture' a larger range of wind directions. These points have now been added to Section 7 of the manuscript (penultimate paragraph).

[Figure]

Figure 5: Left: Empirical posterior predictive intervals (mean $\pm$ 1SD) for the differences between Gaussian plume dispersion model predicted values and observed enhancements, using the 20% of EC Tower data during the 5.8 g min$^{-1}$ release-rate period, plotted by wind direction. Those intervals in red do not contain zero. Right: The same plot zoomed in on wind directions between 280 and 360 degrees.

**It would also be an option to use data that are not used in the inversion for evaluating the optimized concentrations.**

This comment concerns issue (iii) raised in the introduction to this review. Controlled release experiments are unique in that the source location and emission rate are known. Since our inferential target in this work is the emission rate, and we can compare our inferences directly to the true value, we do not provide an analysis on the optimised concentrations in the manuscript. However, in response to this comment, we have now re-run the inversion on 80% (randomly selected) of the observations from the EC Towers during the 5.8 g min$^{-1}$ release-rate period. We then sampled 100 values for $\omega_y$, $\omega_z$, $\boldsymbol{\tau}$, and $Q$ from the MCMC trace plots, and used these to compute empirical posterior predictive intervals (mean $\pm$ 1SD) for the difference between Gaussian plume model predicted concentrations and the observed enhancements on the remaining 20% of EC Tower data during this release-rate period (Gelman et al., 2014, Chapter 6). Figure 5 shows these posterior predictive intervals plotted by wind direction. Those intervals which do not contain zero (which would suggest a poor fit) are plotted in red, while those that do contain zero are plotted in black. Note how the intervals widely vary in size, due to the heteroscedasticity introduced in the measurement-error model that accounts for wind speed and stability class. Of the 594 intervals, 526 ($\approx$ 89%) contain zero. When the intervals are extended to $\pm$ 2SD, 578 ($\approx$ 97%) contain zero. Considering the practical difficulty of exactly quantifying uncertainty in these applications, these values compare reasonably well to what we would expect theoretically (68% and 95%, respectively).

**2.2 Specific comments**

**Page 1: Introduction section: Here I was missing some information about the specific application of inverse modelling that is studied here, among the large range of applications discussed in literature. Special for this case is the small scale of the experiment and the known location of the emission source. Usually this is not the case, raising the question for which kind of application this would be relevant (you might argue that there are easier methods to monitor emissions when the source**

is known).

This comment concerns issue (i) raised in the introduction to this review. Please see our reply to Reviewer 1's comment on this same point on Pg. 4 of this response.

**Page 6, line 22: '... serves to scale outputs vertically ...' In the end I understood that 'vertically' referred to the y-axis in figure 2. What is meant is that Q scales the concentration enhancements. Please rephrase to make this clearer.**

We have now rephrased the sentence to indicate that by scaling we mean multiplying by a constant factor.

**Page 7, line 18: I'm missing an explanation of the logic here. Please clarify the reason for quantifying the statistics of 1/U.**

From Eq. (1) in the manuscript we see that, when conditioned on all other values, the Gaussian plume dispersion model is proportional to the inverse of the wind speed, and hence, the variance of a predicted concentration is proportional to the variance of the inverse of the wind speed. We use this relationship to establish a wind-speed dependent variance for the concentrations. We have expanded the explanation in Sect. 3.2 to make this clearer.

**Page 9, line 7: '... graphically in Fig. 3 ...' Here the dependent variables of the optimization are introduced, but not explained. Here the meaning of tau and the omega's should be explained explicitly, and that in addition to these variables Q is estimated from the data.**

These variables are explained later on in the text while the diagram is there to guide the reader through the subsequent sections. We have now given a brief definition of the symbols upon their first appearance in Sect. 4, and noted the section numbers in which they are each described in more detail.

**Page 10, line 2: '... and one to the stability class ...'. The model error contribution to e_i is not just the stability class.**

This statement is not claiming that the model error contribution is just the stability class, rather that the uncertainty that we are going to associate with each model prediction is stability-class (and instrument-type) dependent. The consequence of this decision is that we cannot use a single variance parameter to explain model-output discrepancy. This is needed since the plume model behaviour changes considerably with stability class. There is also other model uncertainty that we incorporate through the use of prior distributions on $\omega_y$ and $\omega_z$. These also help to describe the transport model uncertainty.

**Page 10, line 3: "(variance)" does not correspond to "(increases)". This is a good example why this grammar style is better avoided.**

We apologise for the lack of clarity. We have now restructured this sentence.

**Page 10: What justifies a windspeed independent error for windspeeds > 1 m/s?**

The Gaussian plume dispersion model is known to be unreliable for low wind speeds. The decision of what constitutes a low wind speed is somewhat subjective and application (scale) dependent, but we see that our results are not too sensitive to the choice of threshold. Further, discarding data at low wind speeds may make the inferential problem harder since this results in less data with which to calibrate the transport model. Please see our reply to Reviewer 1 on

Pg. 2 and Figs. 1 and 2 in this response for more details.

**Page 11, line 7: Is a 'point mass at zero' not just a 'point emission of zero'. If so then please change to avoid confusion.**

The "spike and slab" distribution is a type of mixed distribution, consisting of a "spike" in the probability density at zero, and another (usually continuous) distribution. In our case, we can place a spike and slab prior distribution on the emission rate by setting the half-normal distribution to be the "slab", and a dirac mass at zero to be the spike. This same prior distribution can be employed irrespective of whether the physical emission source is a single point or an area/region. More details of this is given below in response to Reviewer 2's comment regarding the 95% posterior credible intervals not containing zero when the source is switched off.

**Page 11, line 21: What is the relevance of the distribution of the inverse of the square root of the precision parameter? For a precision it would be straightforward to relate it to the numbers that are given in the same sentence. However, what is done here is more complicated for a reason that I dont see.**

The precision of the measurement error is the inverse of the squared of its standard deviation. The standard deviation is on the same scale, and has the same units, as the concentrations. Thus it is more interpretable than the precision. Precisions or variances are however useful when doing MCMC as they allow for conjugate priors to be used, which simplifies and potentially speeds up the sampling process. This is the reason why we tune our prior distributions on the standard deviation rather than directly on the precision, but use precision within the MCMC framework. We now explicitly define the precision as the inverse variance (as per its conventional definition in statistics) before Eq. (3).

**Page 11, line 7: 'While addressing ... zero emission rate'. Looking at equation 4, I don't see why it precludes zero as it is in the interval where the half-normal prior applies.**

This comment concerns issue (iii) raised in the introduction to this review. The lower bound on the support of the half-normal distribution is zero, which means that by construction it is not possible for a 95% posterior credible interval to contain zero. As such, 2.5% of the posterior estimates must lie between zero and the lower limit of the 95% posterior interval, thus precluding zero in the interval.

In the manuscript we suggested that the use of a "spike and slab" prior may be a solution to this problem. We give the mathematical details required to incorporate the spike and slab prior in our framework in an Appendix to this reply.

We have now run four additional inversions using the spike-and-slab prior distribution: one for each release rate using observations collected when the methane source was switched on, and one for each release rate using observations collected when the methane source was switched off. These inversions were run using the EC Tower data in all instances. The results are shown in Fig. 6 of this response. The plots are the same as those given in the Results section of the manuscript, however to accommodate the "spike" portion of the distribution, there is now a circle plotted at zero. The radius of this circle is proportional to the posterior probability that the source on/off variable, $Z_Q$, is equal to zero. Here the results show no change in the posterior credible intervals for the case when the source is switched on, however when the source was switched off, the results now clearly suggest that the emission rate is zero, giving a posterior probability that $Z_Q = 0$ of more than 0.99 for both release-rate periods. The spike and slab

[Figure]

Figure 6: Left: 95% posterior credible intervals for the emission rate $Q$ when the methane source is switched on using the EC Tower data, and in each release rate period: The 5.8 g min$^{-1}$ interval is shown in red, and the 5.0 g min$^{-1}$ interval is shown in blue. The vertical dashed lines represent the true emission rate in each case. The radius of the circle at $Q = 0$ is proportional to the posterior probability that the source on/off variable, $Z_Q$, is equal to zero. Right: Same as left but when the methane source is switched off.

prior thus can give a posterior inference of zero emission rate, as desired. We caution that what we show here is a (promising, but) preliminary result given the allowed time frame for the response, and will be investigated in more depth in future work.

**Page 14, figure 5: It is explained in the text why the method precludes zero as a solution to the inverse problem (see my earlier comment on that). However, I had expected the estimates to be much closer to zero when emissions are switched off. The likely reason is not the zero condition, but the accuracy at which the background concentration is accounted for. It makes me wonder why the background is not fitted as an additional unknown parameter.**

This comment concerns issue (iv) raised in the introduction to this review. We have seen in previous works (e.g., Zammit-Mangion et al., 2015), as well as our own experimentation with background estimation, that the 5th-percentile (by instrument type) is a relatively robust estimator for the background concentration. While the implementation of the spike and slab distribution above appears to have resolved the issue of the 95% posterior credible intervals not containing a zero emission rate, the background can be fitted as a collection of unknown parameters (one for each instrument type) if needed/desired, within the Bayesian framework. In Sect. 4.1 we now refer to Ganesan et al. (2015) who shows how this can be done in a Bayesian setting.

**Page 15, line 23: why are wind speed and direction assumed to be known?**

Due to the small study region and our own instruments recording the weather "on site" throughout the experiment, we assume that the wind speed and direction measurements are of good quality. Wind speed and direction were measured at EC.A using both 2D and 3D instruments. The 3D measurements were obtained from a Campbell CSAT3 sonic anemometer, and the 2D measurements from a Gill WindSonic (Both sonic anemometers were using factory calibration). As part of data quality control, horizontal wind speed and wind direction data from the two instruments were compared, with no arising issues. Wind directions were determined by manually aligning the sonic anemometers so that the reference direction was true north. Before averaging over five-minute intervals, the 3D sonic was logged at 10 Hz, and the 2D sonic at 1 Hz. We have now added more information about how the wind speed and wind direction were measured to Sect. 2 of the manuscript (first paragraph).

**2.3  Technical Corrections**

**Page 3, line 6: 'Houweling' instead of 'Houwelling'**

Thank you, this error has been corrected.

**References**

[revised manuscript text omitted]

**Appendix**

In this appendix we give the mathematical details for the inclusion of the spike-and-slab distribution into our inversion framework. First, let $Z_Q$ be an indicator variable identifying whether the source is switched off ($Z_Q = 0$) or on ($Z_Q = 1$). Then, we can write the observed concentrations as

$$C_i = Z_Q Q \tilde{C}_i,$$

where

$$\tilde{C}_i = \frac{1}{2\pi U_i \tilde{\sigma}_{y_i,k_i} \tilde{\sigma}_{z_i,k_i}} \exp\left(-\frac{y_i^2}{2\tilde{\sigma}_{y_i,k_i}^2}\right) \left[\exp\left(-\frac{(z_i - H)^2}{2\tilde{\sigma}_{z_i,k_i}^2}\right) + \exp\left(-\frac{(z_i + H)^2}{2\tilde{\sigma}_{z_i,k_i}^2}\right)\right].$$

The spike and slab prior distribution is defined through our prior distributions on $Q$ and $Z_Q$ as follows:

$$p(Q) = \begin{cases} \frac{\sqrt{2}}{\sigma_Q \sqrt{\pi}} \exp\left(-\frac{Q^2}{2\sigma_Q^2}\right), & Q \in [0, \infty) \\ 0, & \text{otherwise}, \end{cases}$$

$$p(Z_Q) = \begin{cases} \pi_Z, & Z_Q = 1, \\ 1 - \pi_Z, & Z_Q = 0, \\ 0, & \text{otherwise}, \end{cases}$$

where $\pi_Q$ is the *prior* probability of a source being on. For clarity, let $\boldsymbol{\psi} \equiv \{\boldsymbol{\tau}, \omega_y, \omega_z, \boldsymbol{U}, H, \boldsymbol{\Theta}\}$. The likelihood function now is

$$p(\boldsymbol{Y} \mid Q, Z_Q, \boldsymbol{\psi}) = \prod_{i=1}^{N} \frac{\sqrt{\tau_{m_i}}}{\sqrt{2\pi}} \exp\left(\frac{-\tau_{m_i}(Y_i - Z_Q Q \tilde{C}_i)^2}{2}\right).$$

The full conditional distribution for $Q$, given by

$$p(Q \mid \boldsymbol{Y}, Z_Q, \boldsymbol{\psi}) \propto p(Q)p(\boldsymbol{Y} \mid Q, Z_Q, \boldsymbol{\psi})$$

$$\propto \exp\left(\frac{-Q^2}{2\sigma_Q^2} - \frac{\sum_{i=1}^{N} \tau_{m_i}(Y_i - Z_Q Q \tilde{C}_i)^2}{2}\right) \prod_{i=1}^{N} \frac{\sqrt{\tau_{m_i}}}{\sqrt{2\pi}},$$

has no known form, so we use a Metropolis proposal to sample from this distribution. That is, at the $t$th iteration of our Gibbs sampler we propose $Q^* \sim \text{Gau}(Q^{t-1}, (\sigma^*)^2)$. where $\sigma^*$ is the standard deviation of the proposal distribution for $Q$. Then, assuming $Q$ is sampled at the end of every Gibbs-sampler iteration, we compute the acceptance ratio via

$$\rho = \frac{p(Q^*)p(\boldsymbol{Y} \mid Q^*, Z_Q^t, \boldsymbol{\tau}^t, \omega_y^t, \omega_z^t, \boldsymbol{U}, H, \boldsymbol{\Theta})}{p(Q^{t-1})p(\boldsymbol{Y} \mid Q^{t-1}, Z_Q^t, \boldsymbol{\tau}^t, \omega_y^t, \omega_z^t, \boldsymbol{U}, H, \boldsymbol{\Theta})},$$

and accept the proposed $Q^*$ with probability

$$\alpha = \min\{\rho, 1\}.$$

That is, we draw $u \sim \text{Unif}[0, 1]$, and set $Q^t = Q^*$ if $\alpha \geq u$. Otherwise we set $Q^t = Q^{t-1}$.

The full conditional distribution for $Z_Q$ is given by

$$p(Z_Q \mid \boldsymbol{Y}, Q, \boldsymbol{\psi}) = \frac{p(Z_Q)p(\boldsymbol{Y} \mid Q, Z_Q, \boldsymbol{\psi})}{p(\boldsymbol{Y} \mid Q, \boldsymbol{\psi})}$$

$$= \frac{p(Z_Q)p(\boldsymbol{Y} \mid Q, Z_Q, \boldsymbol{\psi})}{p(Z_Q = 0)p(\boldsymbol{Y} \mid Q, Z_Q = 0, \boldsymbol{\psi}) + p(Z_Q = 1)p(\boldsymbol{Y} \mid Q, Z_Q = 1, \boldsymbol{\psi})}.$$

Now, for $Z_Q = 1$,

$$p(Z_Q = 1)p(\boldsymbol{Y} \mid Q, Z_Q = 1, \boldsymbol{\psi}) = \pi_Z \exp\left(\frac{-\sum_{i=1}^{N} \tau_{m_i}(Y_i - Q\tilde{C}_i)^2}{2}\right) \prod_{i=1}^{N} \frac{\sqrt{\tau_{m_i}}}{\sqrt{2\pi}}$$

$$= P_1,$$

and, for $Z_Q = 0$,

$$p(Z_Q = 0)p(\boldsymbol{Y} \mid Q, Z_Q = 0, \boldsymbol{\psi}) = (1 - \pi_Z) \exp\left(\frac{-\sum_{i=1}^{N} \tau_{m_i} Y_i^2}{2}\right) \prod_{i=1}^{N} \frac{\sqrt{\tau_{m_i}}}{\sqrt{2\pi}}$$

$$= P_0.$$

Finally, we therefore have that

$$p(Z_Q \mid \boldsymbol{Y}, Q, \boldsymbol{\psi}) = \begin{cases} \frac{P_1}{P_0 + P_1}, & Z_Q = 1, \\ \frac{P_0}{P_0 + P_1}, & Z_Q = 0, \end{cases}$$

$$= \begin{cases} \pi_Z^*, & Z_Q = 1, \\ 1 - \pi_Z^*, & Z_Q = 0, \end{cases}$$

$$\sim \text{Bern}(\pi_Z^*),$$

where $\pi_Z^*$ is the *posterior* probability that the source is switched on. This distribution can be sampled from directly without the need for a Metropolis step in our Gibbs sampler.